# The Arabidopsis Deubiquitylase OTU5 Suppresses Flowering by Histone Modification-Mediated Activation of the Major Flowering Repressors *FLC*, *MAF4*, and *MAF5*

**DOI:** 10.3390/ijms24076176

**Published:** 2023-03-24

**Authors:** Ramalingam Radjacommare, Shih-Yun Lin, Raju Usharani, Wen-Dar Lin, Guang-Yuh Jauh, Wolfgang Schmidt, Hongyong Fu

**Affiliations:** Institute of Plant and Microbial Biology, Academia Sinica, Taipei 115, Taiwan

**Keywords:** OTU, deubiquitylation, flowering, epigenetics, *FLC*, *MAF4*, *MAF5*, ARP6, HUB1, SWR1c

## Abstract

Distinct phylogeny and substrate specificities suggest that 12 Arabidopsis Ovarian Tumor domain-containing (OTU) deubiquitinases participate in conserved or plant-specific functions. The *otu5-1* null mutant displayed a pleiotropic phenotype, including early flowering, mimicking that of mutants harboring defects in subunits (e.g., ARP6) of the SWR1 complex (SWR1c) involved in histone H2A.Z deposition. Transcriptome and RT-qPCR analyses suggest that downregulated *FLC* and *MAF4-5* are responsible for the early flowering of *otu5-1*. qChIP analyses revealed a reduction and increase in activating and repressive histone marks, respectively, on *FLC* and *MAF4-5* in *otu5-1*. Subcellular fractionation, GFP-fusion expression, and MNase treatment of chromatin showed that OTU5 is nucleus-enriched and chromatin-associated. Moreover, OTU5 was found to be associated with *FLC* and *MAF4-5*. The OTU5-associated protein complex(es) appears to be distinct from SWR1c, as the molecular weights of OTU5 complex(es) were unaltered in *arp6-1* plants. Furthermore, the *otu5-1 arp6-1* double mutant exhibited synergistic phenotypes, and H2A.Z levels on *FLC/MAF4-5* were reduced in *arp6-1* but not *otu5-1*. Our results support the proposition that Arabidopsis OTU5, acting independently of SWR1c, suppresses flowering by activating *FLC* and *MAF4-5* through histone modification. Double-mutant analyses also indicate that *OTU5* acts independently of the HUB1-mediated pathway, but it is partially required for *FLC*-mediated flowering suppression in autonomous pathway mutants and *FRIGIDA*-Col.

## 1. Introduction

Deubiquitylation enzymes (DUBs), belonging to seven phylogenetically unrelated families, carry distinct and overlapping biochemical activities, including processing ubiquitin (UB) precursors, adducts, and unanchored chains, recycling UB during proteasomal proteolysis, and erasing/editing ubiquitylation on substrates [1,2]. Six DUB classes are cysteine proteases, including UB C-terminal hydrolases (UCH), UB-specific processing proteases (UBP/USP), Ovarian Tumor-related (OTU) proteases, Josephin/Machado–Joseph disease proteases (MJD), MINDYs (MIU-containing novel DUB family), and ZUP1 (Zinc finger containing Ub Peptidase 1). The remaining DUB class covers the JAMM (JAB1/MPN/Mov34)-domain containing and zinc-dependent metalloproteases.

DUBs of different classes have been extensively elucidated, particularly in yeast (*Saccharomyces cerevisiae*) and mammals, for their pivotal roles in various cellular processes and biochemical characteristics in terms of structures, catalytic mechanisms, and substrate specificities [3]. Deubiquitylation, catalyzed by DUBs of various classes, has been firmly established to be mechanistically critical for numerous cellular activities, including 26S proteasome-mediated proteolysis, endocytosis, chromatin silencing, transcriptional activation, splicing and export of mRNAs, cell division, DNA damage response, and signal transduction [3].

The ovarian tumor domain (OTU) was first found with the *D. melanogaster Ovarian Tumor* gene product [4] and subsequently detected in proteins from various eukaryotes (both animals and plants), viruses, and a single bacterial representative, *Chlamydia pneumonia* [5]. Probed by UB derivatives with thiol-reactive C-terminal groups, the human (*Homo sapiens*) OTU domain-containing protein otubain 1 (OTUB1) was identified as a novel deubiquitinase [6]. Sixteen OTU DUBs were subsequently discovered from *H. sapiens*, and most of them have been thoroughly analyzed biochemically and functionally [7]. Unlike UCH and UBP/USP DUBs, each of the human OTU DUBs displays a distinct preference for specific linkage types [8]. Based on phylogenetic analysis, seven human OTU DUBs belong to five cross-kingdom conserved clades with single or dual members: OTUB1/OTUB2, OTUD5, OTUD3, OTUD6A/OTUD6B, and YOD1 (OTUD2). The remaining nine human OTU DUBs were classified as human-specific and consisted of two single-member clades, OTUD1 and OTULIN (FAM105B), one dual-member clade, OTUD4/ALG13, and one five-member A20-like clade, VCPIP1 (VCIP135)/ZRANB1 (TRABID)/A20 (TNFAIP3)/OTUD7A (Cezanne2)/OTUD7B (Cezanne) [9]. Functional analyses have revealed the pivotal roles of human OTU DUBs in central cellular processes or mammalian-specific physiology [7]. The roles and mechanisms elucidated for human OTU DUBs are instructive for investigating OTU DUBs, in particular corresponding orthologues, in other species.

Involvement in central cellular processes is associated with three human OTU members of cross-kingdom conserved clades. OTUB1 and OTUB2 (~62% similar) exhibit distinct linkage specificities, and only OTUB1 acquires binding affinity toward specific E2s such as Ube2N (Ubc13) and Ube2D (UbcH5) to inhibit, in a noncatalytic way, their conjugation activities [10]. Both otubains are involved in the RNF8- and RNF168-mediated DNA-damage response (DDR) pathways. While OTUB2 plays a negative role in removing RNF8-mediated K63-linked ubiquitylation on an early pathway component [11], OTUB1 inhibits RNF168-mediated K63-linked ubiquitylation on DNA damage sites in the late phase of DDR by binding and prohibiting Ube2N-Ube2V catalysis [12,13]. In association with p97/Cdc48, YOD1 (OTUD2) is involved in endoplasmic-reticulum-associated protein degradation (ERAD) and the clearance of damaged lysosomes by macroautophagy [14,15]. No such involvement in central cellular processes was observed for the remaining cross-kingdom-conserved human OTU DUBs. However, their functional importance in vivo has also been extensively demonstrated [16,17].

Human OTU DUBs that are not cross-kingdom conserved also appear to be functionally important in vivo. A few interesting examples are described. OTUD4 targets K63-linked ubiquitin signals marked on receptor-associated factor MyD88 to negatively regulate IL-1 and Toll-like receptor-mediated inflammatory signaling and NF-κB activation [18]. OTUD1 plays a positive role in antifungal innate immunity by deubiquitylating and stabilizing the adaptor of C-type lecti`n receptors to activate canonical NF-κB and MAPK pathways [19]. In contrast, OTUD1 negatively regulates innate immune signaling by deubiquitylating the K6- or K63-linked ubiquitin chain on the regulator IRF3 to modulate its binding to target genes and nuclear translocation [20,21]. OTULINE is the only exclusive M1-linkage-specific DUB associated with the M1-specific E3 LUBAC and plays a pivotal regulatory role in various NFκB and cell death signaling pathways, interferon/antiviral signaling, xenophagic clearance of intracellular bacteria, and proteasomal degradation of cellular protein aggregates [22,23]. As a representative member of the A20-clade, A20 has an anti-inflammatory function by directly inhibiting diverse NF-κB pathways, and indirectly by inhibiting various forms of programmed cell death [24,25]. Also, VCIP135/VCPIP1 is required for the p97-p47- and p97-p37-mediated membrane fusion pathways critical for the reassembly of the ER network and Golgi apparatus during the cell cycle [26]. Of particular interest, TRABID is involved in TLR-induced epigenetic activation by demethylation of suppressive histone marks on the promoters of *Il12* and likely *Il23* encoding the cytokines interleukin 12 (IL12) and IL23 critical for innate and adaptive immune responses via removal of K29- and K11-linked ubiquitin chains assembled on a histone demethylase Jmjd2d to reduce its turnover in cells of the innate immune system, [27].

Database searches identified ~60 Arabidopsis (*Arabidopsis thaliana*) DUBs [9,28], including 3 UCHs, 27 UBPs (grouped into 14 families), 12 OTU proteins, 3 MJDs, 5 JAMNs, 3 MINDYs (At4G11860, At4G22960, and At1g43690), and 2 ZUP1 (At5G24680 and At3G48380). Recent studies with null mutants have revealed the importance of Arabidopsis DUBs of various classes in growth and development, and in adaptive responses [29]. The most extensively investigated plant DUBs belong to the UBP family, with single- or multiple-member phylogenic clades [28,29].

The 12 Arabidopsis OTU proteins display distinct phylogenetic relationships and biochemical properties, and are likely involved in both cross-kingdom conserved cellular activities and plant-specific functions [9]. Three Arabidopsis OTU members from cross-kingdom conserved clades have been further investigated. In association with histone demethylase KDM1C, OTLD1 is likely involved in the repression of specific target genes associated with plant growth by the removal of various histone activation marks, such as H2B monoubiquitination, H3 acetylation, and trimethylation of H3K4 [30,31]. Interestingly, OTLD1 can also target and activate specific genes by H2B deubiquitination to enhance the accumulation of active histone marks, such as H3 acetylation and H3K4 trimethylation, or in cooperation with KDM1C, remove the repressive mark H3K9me2 [32,33]. However, as T-DNA insertion mutants did not show obvious defects, the in vivo functional roles of Arabidopsis OTLD1 have yet to be demonstrated. The rice genome harbors 20 *OTU* loci [34]. Interestingly, together with HUB2, the rice OTLD1 orthologue (OsOTU2) plays a regulatory role in ABA signaling and drought resistance by modulating H2B ubiquitination. OsOTU2 is recruited by the ABA signaling suppressor MODD to suppress the expression of bZIP46 target genes involved in ABA signaling and drought resistance by reducing H2Bub1 levels [35].

Surprisingly, as a potential human OTUB1 orthologue involved in the DNA damage response [12,13], Arabidopsis OTU1 was reported to be an ERAD component [36]. Homozygous *otu1* mutant plants showed slightly increased sensitivity to salt and common ER stress inducers, as well as altered turnover rates of artificial ERAD markers in *N. benthamiana* leaves when co-infiltrated with either wild-type or catalytic inactive OTU1 variants [36]. However, due to detected binding to UBC13 and inhibition of UBC13/UEV1b-mediated K63-linked ubiquitin chain assembly [36], a potential role of Arabidopsis OTU1 in DNA damage response cannot be ruled out. In contrast, functional roles have not been assessed for Arabidopsis OTU2, the potential orthologue of yeast YOD1, and mammalian OTUD2, which are involved in ERAD [14,37]. Interestingly, OTU1 also plays a role in controlling seed/organ size through epigenetic regulation of *DA1* and *DA2*, repressors involved in seed and organ size control [38]. Arabidopsis defective in the expression of *OTU1* showed increased histone H2B ubiquitination and simultaneously elevated euchromatic histone marks H3K4me3 and H3 acetylation on the *DA1* and *DA2* loci.

Distinct root growth phenotypes were observed in *otu5* mutant plants under phosphate (Pi)-replete or -deficient conditions, suggesting that Arabidopsis OTU5 plays a role in root growth in response to phosphate [39,40]. Under Pi-replete conditions, *otu5* plants displayed reduced primary root growth with increased root hair length and density, phenotypes that resembled Pi-deficient plants. The *otu5* root growth phenotypes mimic those of *arp6*, a mutant defective in a subunit of the SWR1 chromatin-remodeling complex (SWR1c), which showed a constitutive phosphate starvation response (PSR) due to compromised H2A.Z deposition on PSR genes [41]. In contrast, under Pi starvation, *out5* plants formed very short primary roots and root hairs. However, OTU5 function is likely distinct from SWR1c-mediated H2A.Z deposition, as phosphate starvation response genes were not activated in the *otu5* mutant (as was the case for *arp6*), and synergistic root growth phenotypes were observed for the *otu5 arp6* double mutant [40].

Multi-omics analyses conducted on *otu5* roots revealed only subtle differences in DNA methylation levels [40,42]. Changes in DNA methylation patterns had very little correlation with gene expression levels and histone methylation peaks (H3K4me3 or H3K27me3) [42]. Thus, the functional implication of OTU5 on specific target genes remains unequivocally demonstrated. Similarly, proteomic analyses of *otu5* plants suggest that the various root growth phenotypes of *otu5* plants are possibly caused by the altered abundance of proteins involved in chromatin organization, root hair morphogenesis, and redox homeostasis [40]. However, the causative mechanisms still need to be elucidated in detail.

To further investigate the mechanistic basis of OTU5 action, we focused on the early flowering phenotype of *otu5* plants. Intriguingly, our results demonstrate that nucleus-localized and chromatin-associated OTU5 participate in flowering suppression by activating *FLC*-related flowering repressors through histone modification. Genetic and biochemical analyses suggest that OTU5 acts independently of ARP6 and HUB1 and is partially required for *FLC*-mediated flowering suppression in autonomous pathway mutants and *FRIGIDA*-Col.

## 2. Results

### 2.1. Pleiotropic Vegetative and Reproductive Growth Phenotypes Are Associated with the otu5 T-DNA Insertion Mutant

Phylogenetic relationships and biochemical properties suggest that Arabidopsis OTU proteins belonging to each of the five cross-kingdom conserved and plant-specific OTU clades could potentially be involved in distinct in vivo functions [9]. As an initial step to examine the specific in vivo functions of 12 *A. thaliana OTU* loci, we established homozygous T-DNA insertion mutants for 8 of them, designated *otu1-1*, *otu2-1*, *otu3-1*, *otu5-1*, *otld1-1*, *otu7-1*, *otu10-1*, and *otu12-1* (Appendix A). Insertion mutants for other *OTU* loci were not available when the project was initiated. Based on the RT-PCR results, *otu1-1* and *otu12-1* showed reduced levels of their corresponding full-length transcripts. However, OTU1 proteins were not detected in *otu1-1* (JKG and HF, unpublished), indicating that *otu1-1* is a knockout mutant or expresses full-length proteins below a detectable level. The remaining *OTU* T-DNA lines are likely knockout mutants or produce truncated transcripts since the corresponding full-length transcripts were not detected (Appendix A). The overall morphology of the established *OTU* T-DNA lines was compared throughout their development with wild-type Col-0 plants. At 14 and 21 days after stratification (DAS), only *otu3-1*, *otu5-1*, and *otu12-1* showed reduced rosette sizes; the remainder of the *OTU* T-DNA lines grew similarly to the wild type (Appendix A). However, at 45 DAS, only *otu5-1* plants showed a clearly reduced final plant height; a near wild-type growth phenotype was observed for the rest of the examined *OTU* mutant lines (Appendix A). Morphometric measurements of mature plants at 45 DAS were conducted for all *OTU* mutants under study. Here, only *otu5-1* revealed clear morphological differences from wild-type plants. Primary and cauline branch inflorescences of *otu5-1* plants were significantly shorter than those of wild-type plants (average: 30.4 ± 2.9 and 26.0 ± 3.1 cm for *otu5-1* (*n* = 24) and 37.3 ± 2.6 and 29.4 ± 1.7 cm for wild-type plants; *n* = 24). Moreover, only *otu5-1* plants flowered significantly earlier under both long- and short-day conditions in comparison with wild-type plants. The average rosette leaf numbers at the time of bolting were 9.2 ± 0.5 (*n* = 13) under long-day conditions and 28.6 ± 5.2 (*n* = 16) under short-day conditions for *otu5-1* plants, in comparison with 12.1 ± 0.6 and 80.1 ± 6.1, respectively, for wild-type plants.

Since obvious growth defects were observed in *otu5-1* plants, we characterized the vegetative and reproductive growth phenotypes for this mutant in more detail. The *otu5-1* mutant was confirmed to be null. Its T-DNA was determined to be located 23 bp downstream and 22 bp upstream of the start codon of *OTU5a* and *OTU5b*, respectively, which apparently disrupted transcription and translation; *OTU5* transcripts and proteins of both isoforms were not detected (Figure 1A, middle and right panels), as previously described [40]. Similarly, as described above, *otu5-1* plants grew at a slower rate and showed markedly reduced rosette sizes at early developmental stages (i.e., at 14 and 21 DAS) when compared with wild-type plants (Figure 1B and Appendix A). Examined at 27 DAS, the number and size of *otu5-1* rosette leaves grown under long-day conditions were reduced in comparison with Col-0 (Figure 1C, left panel). Significantly reduced surface areas were detected for the fifth and above rosette leaves (Figure 1C, right panel). The *otu5-1* plants displayed a significantly reduced final plant height at 45 DAS (Figure 1D and Appendix A). The primary inflorescence stem of *otu5-1* plants was much thinner than that of wild-type plants (Figure 1E). In agreement with previous reports [39,40], reduced primary root lengths and increased root hair density and length were observed for *otu5-1* seedlings grown under phosphate-replete conditions compared to wild-type plants (Figure 1F,G).

The *otu5-1* plants flowered significantly earlier than the wild type under both long- and short-day conditions when measured either by rosette leaf number or DAS at bolting (Figure 2A, left and right panels, and described above). In addition to rosette leaves and inflorescence stems, reduced sizes were also observed in the various floral organs of *otu5-1* mutant plants. In comparison to Col-0, *otu5-1* flowers appeared to be smaller (Figure 2B), with significantly reduced average surface areas of sepals and petals (Appendix A), as well as reduced average size and length of gynoecia/pistils (Figure 2C and Appendix A) in *otu5-1* plants. The reduced sizes of various *otu5-1* organs are likely primarily caused by reduced cell numbers, as similar densities of epidermal cells and stomata in rosette leaves were observed in *otu5-1* and wild-type plants (Appendix A). The cell density of adaxial epidermal cells for *otu5-1* petals was, however, slightly increased, suggesting that, in addition to reduced cell numbers, reduced petal cell sizes are responsible for the reduced *otu5-1* petal size (Appendix A).

The average silique length of *otu5-1* was significantly reduced compared to Col-0 (Figure 2D), indicating a possible partial fertility defect. Besides the reduced average ovule number per silique for *otu5-1* (Figure 2E), on average, 8% of the ovules from the *otu5-1* siliques were aborted (Figure 2F, left and right panels). As determined by reciprocal crosses between *otu5-1* and Col-0, fertility defects appear to be largely associated with defective female gamete development. A significant and similar reduction in seed number was observed in siliques of *otu5-1*-selfed and *otu5-1*/Col-0-crossed but not in Col-0/Col-0- and Col-0/*otu5-1*-crossed compared with Col-0-selfed individuals (Figure 2G).

To verify that the growth phenotypes associated with *otu5-1* plants are caused by disruption of the *OTU5* locus, we complemented the *otu5-1* plants using an N-terminal GS-tagged OTU5a construct (*GS-OTU5a*) driven by the CaMV 35S promoter, a construct established for future purification of the OTU5-containing complex. Multiple complementation lines were generated that expressed transgenic GS-OTU5a proteins at significantly higher levels than endogenous OTU5a and OTU5b in Col-0 (Figure 1A, right panel, Complement *Wt-1* and *Wt-2*). All vegetative *otu5-1* growth phenotypes appeared to be rescued when *GS-OTU5a* was introduced. Exemplified by two complemented *otu5-1* lines (Figure 1, complement *Wt-1* and *Wt-2*), the examined vegetative growth phenotypes, including rosette size of seedlings at 14 DAS (Figure 1B), rosette leaf number and size at 27 DAS (Figure 1C, left and right panels, *Wt-1*), final plant height at 45 DAS (Figure 1D), average primary root length (Figure 1F), and root hair length and density (Figure 1G) were rescued and reverted to those of Col-0 plants.

Similarly, all observed reproductive growth phenotypes of *otu5-1* plants were rescued when *GS-OTU5a* was introduced (Figure 2 and Appendix A, complement *Wt-1* and *Wt-2*). Reproductive growth phenotypes examined included flowering time under long-day conditions, measured either by rosette leaf number or DAS at bolting (Figure 2A, left and right panels), flower size (Figure 2B), average surface areas of sepals and petals (Appendix A), the cell density of adaxial epidermal cells of petals (Appendix A, *Wt-1* and *Wt-2*), size, and length of gynoecia/pistils (Figure 2C and Appendix A), and ovule number and ovule abortion rate (Figure 2E,F, right panel).

### 2.2. The Conserved Catalytic Site of Arabidopsis OTU5 Is Essential In Vivo

Tested in vitro, OTU5 was found to be inactive in deubiquitylation assays [9]. To examine whether catalytic activity is essential in vivo, we conducted *otu5-1* complementation experiments using an N-terminal GS-tagged OTU5a construct, *GS-OTU5a*-*CS*, which encoded a point mutation at the conserved catalytic site (C192S). Complemented *otu5-1* lines harboring *GS-OTU5a*-*CS* expressed the corresponding C192S variant at significantly higher levels than endogenous OTU5a and OTU5b in Col-0 (Figure 1A, right panel, Complement *CS-1* and *CS-2*). Not surprisingly, all examined vegetative growth phenotypes of *otu5-1* harboring *GS-OTU5a*-*CS* (Figure 1, Complement *CS-1* and *CS-2*), including rosette size of seedlings at 14 DAS (Figure 1B), final plant height at 45 DAS (Figure 1D), and average primary root length (Figure 1F), were not rescued and remained similar to those of *otu5-1*. Similarly, all examined reproductive growth phenotypes of *otu5-1* harboring *GS-OTU5a*-*CS* (Figure 2 and Appendix A, Complement *CS-1* and *CS-2*), including flowering time under long-day conditions measured either by rosette leaf number or DAS at bolting (Figure 2A, left and right panels), flower size (Figure 2B), average surface areas of sepals and petals (Appendix A), and size and length of gynoecia/pistils (Figure 2C and Appendix A), were also not rescued and remained similar to those of *otu5-1*. Together, these results indicate that the catalytic activity of OTU5 is essential for in vivo functions.

To further examine the in vivo importance of the conserved catalytic site of OTU5, we generated Col-0 lines overexpressing the same N-terminal GS-tagged wild-type OTU5a (*GS-OTU5a*) or OTU5a-C192S (*GS-OTU5a*-*CS*) used for the complementation experiments. Similar to what was observed in the complementation lines, Col-0 lines harboring *GS-OTU5a* or *GS-OTU5a*-*CS* expressed the corresponding proteins at levels significantly higher than endogenous OTU5a and OTU5b in Col-0 (Figure 1A, right panel, overexpressing *Wt-1*, *Wt-2*, *CS-1*, and *CS-2*). A dominant negative effect was expected for overexpressing C192S but not the wild-type OTU5a variant if the endogenous wild-type OTU5 function competed or interfered with by the catalytic site-mutated OTU5. As expected, vegetative and reproduction growth phenotypes mimicking *otu5-1* were only observed in Col-0 lines overexpressing GS-OTU5a-CS but not in lines overexpressing GS-OTU5a (Figure 1, Figure 2 and Appendix A, overexpressing *CS-1* and *CS-2* versus *Wt-1 and Wt-2*). The *otu5-1*-mimicking phenotypes observed in Col-0 lines overexpressing GS-OTU5a-CS included rosette size of seedlings at 14 DAS (Figure 1B), final plant height at 45 DAS (Figure 1D), slightly early flowing as measured either by rosette leaf number or DAS at bolting (Figure 2A, left and right panels), flower size (Figure 2B), average surface areas of sepals and petals (Appendix A), and size and length of gynoecia/pistils (Figure 2C and Appendix A). However, reduced primary root length mimicking *otu5-1* was not observed in Col-0 lines harboring *GS-OTU5a*-*CS* or *GS-OTU5a* (Figure 1F).

### 2.3. Gene Ontology Enrichment Identified Differentially Expressed Genes Potentially Responsible for the otu5-1 Phenotypes

To determine the molecular mechanism(s) underlying the various *otu5-1* phenotypes, genes that were differentially expressed between *otu5-1* and wild-type plants were identified by comparing transcriptome profiles between 10-day-old *otu5-1* and Col-0 seedlings collected in the morning using two color-based *A. thaliana* genome 44K oligo microarrays. Seventy-one downregulated and sixty-seven upregulated genes with a fold change of ≥2 (*p* < 0.05) were detected in *otu5-1* (Appendix A). Gene Ontology (GO) enrichment of the differentially expressed genes (DEGs) was performed based on the GO annotations of the TAIR10 genome release [43,44]. The significantly enriched GO terms were related to responses to various biotic and abiotic stresses and flower development (Appendix A). Using a *p*-value of <1 × 10^−03^ as a cutoff point for the enriched GO terms of all categories, including flower development (Appendix A), the number of downregulated genes outscored the upregulated genes (36 vs. 7). Downregulated genes associated with flower development included *FLC* (At5g10140) and *MADS AFFECTING FLOWERING 5* (*MAF5*, At5g65080). Interestingly, four of the seven upregulated genes associated with enriched GO terms were annotated as being involved in flower development. The upregulated gene *AGAMOUS-LIKE 20/SUPPRESOR OF OVEREXPRESSION OF CO 1* (*AGL20*/*SOC1*, At2g45660) is of particular interest. The skewed appearance of downregulated genes over upregulated genes associated with enriched GO terms suggests that downregulated genes in *otu5-1* likely contribute more than upregulated genes to the observed *otu5-1* phenotypes.

### 2.4. A. thaliana OTU5 Is Involved in the Epigenetic Regulation of FLC, MAF4, and MAF5

The transcriptome analysis strongly suggested that the early flowering phenotype of *otu5-1* was primarily caused by the downregulation of the flowering suppressor *FLC* and its homolog *MAF5* and the upregulation of the flowering activator *SOC1*. The altered expression of *FLC*, *MAF5*, and *SOC1* was confirmed by RT-PCR and RT-qPCR (Figure 3). On the other hand, the expression levels for various autonomous pathway components, including *FVE* (At2g19520), *FLOWERING LOCUS D* (*FLD*, At3g10390), *FPA* (At2g43410), *FLOWERING CONTROL LOCUS A* (*FCA*, At4g16280), *LUMINIDEPENDENS* (*LD*, At4g02560), and *FY* (At5g13480), were not affected in *otu5-1* (Figure 3A), expression levels of *FLC* and three *MAF* loci (At1g77080, At5g65070, and At5g65080) were significantly downregulated (Figure 3B). Moreover, expression levels of the flowering activator *SOC1* and *FLOWERING LOCUS T* (*FT*, At1g65480) but not the photoperiodic flowering component *CONSTANS* (*CO*, At5g15840) were upregulated in *otu5-1* plants (Figure 3C). The early flowering phenotype and the associated downregulation of *FLC* and *MAF loci* in *otu5-1* plants suggest that OTU5 is putatively involved in the activation of *FLC* and *MAFs*, as well as in flowering suppression.

One potential proposition is that functional OTU5 is critical for the proper expression of some of the differentially expressed genes, especially those associated with flower development. Since *FLC* and *MAF* loci are subjected to epigenetic regulation [45], an even more interesting proposition is that OTU5 directly or indirectly affects the expression of *FLC* and *MAFs* by epigenetic regulation. However, the levels of major histone marks, including H3K4me3, H3K36me3, H3K36me1, H3K27me3, and ubH2B, were not altered in *otu5-1* (Figure 4A), suggesting that OTU5 is not directly involved in the regulation of global histone modification.

To examine the potential involvement of OTU5 in the epigenetic regulation of *FLC* and *MAF4-5*, chromatin immunoprecipitation, in conjunction with qPCR (qChIP), using antisera against various histone marks, was used to determine whether the levels of activating and repressive histone marks on the *FLC*, *MAF4*, and *MAF5* loci were altered in *otu5-1* plants. Because histone H2B deubiquitylation is required for *FLC* activation and flowering suppression [45], we also examined the ubH2B profiles of these flowering repressor loci. Nine to ten regions in various exons and introns from the *FLC* locus covering the full coding region were examined for altered levels of histone marks in *otu5-1* plants (Figure 4B, primer pairs 1–10). As predicted, the levels of activating histone mark H3K36me3 were generally reduced in all tested *FLC* regions in *otu5-1*, as exemplified by the reduced qChIP yields compared with those of the Col-0 plants (Figure 4C, primer pairs 1–9, α-H3K36me3). In contrast, the levels of the repressive histone marks H3K27me3 and H3K9me2 were generally increased across all tested *FLC* regions in *otu5-1* (Figure 4C, primer pairs 1–10 or 1–9 for α-H3K27me3 and α-H3K9me2, respectively). The levels of ubH2B also accumulated in all tested *FLC* regions in *otu5-1* plants (Figure 4C, primer pairs 1–9, α-ubH2B).

The levels of activating and repressive histone marks on *MAF4* and *MAF5* in *otu5-1* were also examined by qChIP with four to six selected regions in various exons and introns covering their full coding regions (Figure 5A, primer pairs 1–6 for *MAF4* and Figure 5B, primer pairs 1–5 for *MAF5*). Slightly reduced levels of the activating histone mark H3K36me3 in two tested regions located in the first introns of *MAF4* (primer pairs 3 and 4) and *MAF5* (primer pairs 2 and 3) were observed in *otu5-1* plants (Figure 5C,D, α-H3K36me3). In contrast, the levels of the repressive histone marks H3K27me3 and H3K9me2 were generally enhanced in all tested regions of *MAF4* and *MAF5* in *otu5-1* plants (Figure 5C,D, α-H3K27me3 and α-H3K9me2). Similar to *FLC*, ubH2B levels accumulated in all tested regions of *MAF4* and *MAF5* in *otu5-1*, except for the region tested with primer pair 2 for *MAF5* (Figure 5C,D, α-ubH2B).

### 2.5. Nuclear OTU5 Is Associated with Chromatin

To be capable of modifying histone marks on *FLC* and *MAF4*-*5*, either directly or indirectly, OTU5 is required to localize to the nucleus. We examined the presence of OTU5 in the total, cytosolic, and nuclear fractions prepared from young seedlings (Figure 6A). Whereas various 26S proteasome subunits, including the core particle subunit PBA1, the base subunits RPT5, RPN10, and RPN13, and the lid subunits RPN5, RPN6, RPN8, and RPN12, as well as the UBA-UBL ubiquitin receptors RAD23s, DDI1, and NUB1 were detected at relatively low levels in the nuclear fraction but at much higher levels in the total and cytosolic fractions; in contrast, both OTU5 isoforms showed clear enrichment in the nuclear fraction but much lower levels in the total and cytosolic fractions. Interestingly, the base subunit RPT2 was also enriched in the nuclear fraction. Unlike OTU5, another OTU member, OTU2, was barely detected in the nuclear fraction but was more abundant in the total and cytosolic fractions. The nuclear enrichment of OTU5 was confirmed by transient expression in *A. thaliana* protoplasts using green fluorescent protein (GFP) fusion constructs (Figure 6B). Both OTU5a and OTU5b were expressed as C- and N-terminally attached GFP fusions. In contrast to the GFP control, which generally showed cytosolic GFP fluorescence, all OTU5 fusions had clear nuclear GFP fluorescence enrichment (Figure 6C). Slight cytosolic and peripheral GFP fluorescence was also detected to various extents for different OTU5 fusions. Whether the peripheral fluorescence originated in the plasma membrane was not determined.

Moreover, to be capable of modifying histone marks on *FLC* and *MAF4*-*5*, OTU5 must be associated with chromatin. As expected, we detected the presence of both OTU5a and OTU5b in the chromatin pellet but not in the nucleoplasmic fraction prepared from isolated nuclei without pre-incubation with micrococcal nuclease (MNase) (Figure 6D, P3− vs. S3−). However, after nuclei pretreatment with MNase to release chromatin-associated proteins, both OTU5a and OTU5b could be detected in the nucleoplasmic fraction but not in nuclear pellets (Figure 6D, S3+ vs. P3+). Our results support the assumption that both OTU5 isoforms are enriched in the nucleus and associated with chromatin.

### 2.6. OTU5 Is Associated with FLC, MAF4, and MAF5 Loci

OTU5 could be a critical mechanistic component involved in histone modification-mediated regulation of specific target genes, such as *FLC*, *MAF4*, and *MAF5*. We examined whether OTU5 was directly associated with the *FLC*, *MAF4*, and *MAF5* loci by qChIP using antisera raised against recombinant OTU5. The same regions tested for various histone marks on *FLC*, *MAF4*, and *MAF5* were tested here (Figure 7A–C, top panels). Referring to input, qChIP products for all tested regions of *FLC*, *MAF4*, and *MAF5* were clearly detected for chromatin isolated from Col-0. In contrast, qChIP products for all tested regions were drastically reduced for chromatin isolated from *otu5-1* (Figure 7A–C, bottom panels). Thus, our results demonstrate that at least a part of chromatin-associated OTU5 is located in *FLC*, *MAF4*, and *MAF5*.

### 2.7. Molecular Weights of the OTU5-Associated Complex(es) Were Unaltered, but Abundance Was Reduced in the Absence of ARP6

Intriguingly, *otu5-1* plants displayed phenotypes very similar to plants carrying a mutation in loci encoding any of the subunits of the SWR1 complex (SWR1c) [46], such as *pie1* [47,48], *arp4* [49], *arp6*/*suf3* [48,50,51], and *swc6/sef* [52,53,54]. The shared phenotypes between *otu5-1* and the SWR1c subunit mutants include *FLC* suppression-mediated early flowering, reduced sizes of various organs, reduced primary root growth, increased root hair density and length, reduced rosette leaf numbers, reduced plant height, and aborted ovules. These phenotypes were also observed in plants carrying *hta9 hta11* double mutations in genes encoding H2A.Z [55]. The phenotypical mimicry of *otu5-1* with mutants in SWR1c subunits and H2A.Z suggests that OTU5 may participate in SWR1c-associated functions, such as chromatin organization [56] or H2A.Z deposition [46]. One intriguing possibility is that OTU5 is a SWR1c-associated factor or even an integrated subunit and thus a mechanistic component of SWR1c-mediated regulation. Moreover, the phenotypic mimicry of *otu5-1* for wild-type plants overexpressing a catalytic site-mutated OTU5 suggests that the mutant proteins likely conferred a dominant negative effect by competing with the endogenous wild-type OTU5 proteins for associated partners or complexes.

As the DUBs of various classes, including OTU domain-containing proteins, often function in complexes [57], one supposition is that OTU5 is a critical component of a protein complex, directly or indirectly involved in histone modification on a subset of specific target genes. To examine whether OTU5 exists in protein complexes, such as SWR1c, extracts of three-week-old Col-0 and *arp6-1* shoots were analyzed by gel filtration chromatography and immunoblotting. For the Col-0 extracts, the majority of the ~34–37 kDa OTU5 isoforms were associated with fractions peaking at ~60 kDa, but significant amounts of both OTU5 isoforms were also associated with fractions peaking at ~500 kDa (Figure 8, Col-0). For *arp6-1* extracts, the abundance of OTU5 isoforms increased with fractions peaking at ~60 kDa, but the abundance of OTU5 isoforms was clearly reduced for fractions peaking at ~500 kDa (Figure 8, *arp6-1*). Since the sizes of the OTU5-associated complex(es) remain unaltered in the absence of ARP6, our results support the assumption that OTU5-associated protein complex(es) are distinct from SWR1c. However, the abundance of the OTU5-associated complex(es) was reduced in the absence of ARP6.

### 2.8. OTU5 Acts Independently of ARP6

To further examine the potential dependence of OTU5 on SWR1c-mediated pathways, we examined the genetic interactions of *OTU5* with *ARP6* (At3g33520). The *otu5 arp6* double mutant was generated by crossing *arp6-1* [51] with *otu5-1* (Figure 9A, *db*). The *otu5 arp6* double mutant clearly showed synergistic effects regarding various vegetative and reproductive growth phenotypes when compared with each of the single mutants (Figure 9). The *otu5 arp6* double mutant further showed a reduced growth rate when compared with each of the single mutants. Whereas each of the single mutants (i.e., *otu5-1* and *arp6-1*) showed similar but smaller rosettes when compared with Col-0, the *otu5 arp6* double mutant displayed a further reduced rosette size at 21 DAS (Figure 9B, *db*). Examined at 27 DAS, *arp6-1* plants displayed smaller and fewer rosette leaves in comparison with *otu5-1*, which in turn had smaller and fewer rosette leaves than the Col-0 plants (Figure 9C). In agreement with a synergistic effect, *otu5 arp6* double-mutant plants displayed a further reduction in the size and number of rosette leaves in comparison with each of the single mutants (Figure 9C, *db*). Moreover, the final averaged plant height of the double mutant was further reduced relative to that of the single mutants at 45 DAS, which already showed reduced plant height in comparison to Col-0 (Figure 9D, *otu5*, *arp6*, and *db*). In agreement with a previous report [40], both *otu5-1* and *arp6-1* single mutants showed reduced primary root growth; significantly reduced primary root growth was also observed for the *otu5 arp6* double mutant in comparison with each of the single mutants (Figure 9E, *db*). In addition, in agreement with previously published results [40], both *otu5-1* and *arp6-1* mutants displayed increased root hair length and density in comparison with Col-0, and a synergistic effect regarding root hair length and density was observed for the *otu5 arp6* double mutant (Figure 9F, *db*).

Synergistic effects on reproductive growth, particularly on fertility, were observed in *otu5 arp6* double-mutant plants in comparison with single mutants. The flower sizes of the single and double mutants were similar and smaller than the wild-type flowers (Figure 9G, left panels). In line with these data, the average sepal and petal areas for the single and double mutants were smaller in comparison with Col-0; however, the average sepal area of *otu5-1* and the average petal area of *arp6-1* were slightly larger than those of the other single mutant or the double mutant (Appendix A). The average stamen lengths for *otu5-1* single mutants and, in particular, *arp6-1* and double mutants were significantly reduced in comparison with Col-0 (Appendix A). In addition, the average pistil length of the single and double mutants was significantly reduced, and the shortest was associated with the double mutant (Figure 9G, right panel and Appendix A). The most drastic synergistic effect was associated with fertility; the average silique length of the *otu5 arp6* double mutant was massively reduced, followed by *arp6-1* and *otu5-1* (Figure 9H,I). Similarly, the smallest average ovule number was observed in *otu5 arp6* double-mutant plants, followed by *arp6-1* and *otu5-1* (Figure 9J, top panel). In agreement with the average silique lengths, the highest ovule abortion rate was observed in *otu5 arp6* double-mutant plants (70%), followed by *arp6-1* (40%), *otu5-1* (8%), and Col-0 (2%) (Figure 9J, bottom panel).

Similar to *otu5-1*, *arp6-1* and *otu5 arp6* showed early flowering under both long-day and short-day conditions, in comparison with Col-0 measured either by DAS or by leaf numbers at bolting (Figure 9K). Under long-day conditions, the *arp6*-*1* plants showed the earliest flowering, followed by *otu5 arp6* double mutant and *otu5-1* (Figure 9K, LD). Under short-day conditions, the earliest flowering was observed in the *otu5 arp6* double mutant and *arp6*-*1* plants, followed by *otu5-1* plants (Figure 9K, SD). Examined by RT-qPCR, *otu5-1*, *arp6-1*, and the *otu5 arp6* double mutant showed significantly reduced expression levels of *FLC* (Figure 9L). Although not exactly correlated with flowering time, the lowest expression levels of *FLC* were observed in the *otu5 arp6* double mutant, followed by *otu5-1* and *arp6-1*. Considering all the results, the increased severity of the *otu5 arp6* double-mutant plants regarding various phenotypes supports the proposition that *OTU5* has a similar in vivo function to *ARP6*. However, the observed synergistic effects indicate that OTU5 acts independently of the ARP6- and likely SWR1c-mediated mechanisms.

### 2.9. Comparison of DEGs in the otu5 arp6 Double Mutant with otu5 and arp6 Single Mutants

To further examine possible mechanistic differences between OTU5- and ARP6-mediated functions, we performed additional transcriptome analyses with 10-day-old seedlings collected in the evening of Col-0, *otu5-1*, *arp6*-1, and *otu5 arp6* using one-color-based Arabidopsis 44K oligo microarrays. By comparing with Col-0, DEGs were identified and analyzed among *otu5-1*, *arp6-1*, and *otu5 arp6* plants. Subsets of 92, 811, and 947 upregulated and 177, 320, and 1578 downregulated genes were identified in *otu5-1*, *arp6*-1, and *otu5 arp6* plants, respectively (Appendix A). More DEGs (both up- and downregulated) were found in *arp6-1* in comparison with *otu5-1* (Appendix A), indicating a stronger impact on the transcriptome when the ARP6 function was lost. Whereas a significant proportion of upregulated (51.1%) and downregulated (18.6%) DEGs found in *otu5-1* overlapped with those of *arp6-1*, only a small number of DEGs found in *arp6-1* were also found in *otu5-1* (Appendix A), suggesting that ARP6 covers most of the OTU5-associated functions. As expected, a significant proportion of the DEGs found in *otu5-1* and *arp6-1* were also found in *otu5 arp6*. However, only a small subset of the DEGs found in *otu5 arp6* were also found in *otu5-1* (Appendix A). In contrast, a relatively large subset of DEGs (mostly upregulated) found in *otu5 arp6* also overlapped with those in *arp6-1*, suggesting that the altered transcriptomic profile of *otu5 arp6* plants was largely due to the lack of functional ARP6.

The expression patterns of the DEGs detected in *otu5-1*, *arp6-1*, and *otu5 arp6* were further analyzed (Appendix A). Interestingly, a relatively larger number of *otu5-1* DEGs, particularly those upregulated, were similarly expressed in *arp6-1* (66.3% upregulated and 28.2% downregulated). In contrast, for *arp6-1*, only 20% of up- and 25.6% of downregulated DEGs showed similar expression changes in *otu5-1*. Moreover, a larger number of *arp6 otu5* DEGs, in particular those upregulated, were similarly expressed in *arp6-1* (80.0% up- and 28.0% downregulated), but a smaller number of *arp6 otu5* DEGs displayed similar expression changes in *otu5-1* (26.1% up- and 12.5% downregulated).

To further investigate the mechanistic differences between OTU5 and ARP6, we examined the potential synergistic effects of 42 up- and 30 downregulated DEGs from the *otu5 arp6* double mutant, which overlapped with those in each of the single mutants. Interestingly, 12 out of 42 (28.6%) upregulated DEGs showed higher expression in *otu5 arp6* when compared with single mutants. The difference in log2-fold changes ranged from 0.23 to 4.2 (Appendix A). More dramatically, 14 out of 30 (46.7%) downregulated *otu5 arp6* DEGs showed more pronounced repression in comparison with single mutants, leading to reduced log2-fold changes between −0.22 to −2.87 compared with single mutants (Appendix A). Of particular interest among the 30 downregulated DEGs, *FLC* (AT5g10140) was detected by three different probes, all of which showed further reduced log2-fold changes in *otu5 arp6* mutant plants, ranging from −0.66 to −1.75 relative to the single mutants (Appendix A).

### 2.10. Levels of H2A.Z on FLC and MAF4-5 Were Not Altered in otu5-1 Plants

To further investigate whether OTU5 acts independently of SWR1c, we compared the H2A.Z levels on *FLC* and *MAF4-5* among *otu5-1*, *arp6-1*, and Col-0 by qCHIP using antisera against the Arabidopsis H2A.Z isoforms HTA9 and HTA11 (α-H2A.Z) using the same set of primers described above (Figure 10A–C, top panels). As expected, H2A.Z levels on *FLC* and *MAF4-5* were reduced in all tested regions in *arp6-1* compared with Col-0. In contrast, in *otu5-1* H2A.Z levels were not altered and comparable to those in Col-0 (Figure 10A–C, bottom panels).

### 2.11. OTU5 Acts Independently of the HUB1-Mediated Flowering Suppression Pathway

Our results showed that increased ubH2B levels on the *FLC* and *MAF4-5* loci were associated with early flowering in *otu5-1*, which is in contrast to what was reported for the *hub1* and *ubc1 ubc2* mutants, in which early flowering was associated with reduced ubH2B levels [58,59]. These conflicting results suggest that OTU5-mediated flowering suppression is distinct from the HUB1-mediated pathway. To confirm this supposition, we generated two double mutants, *otu5 hub1-4* and *otu5 hub1-5*, by crossing *otu5-1* separately into the *hub1* alleles *hub1-4* and *hub1-5*, respectively. We then compared the flowering times of the single and double mutants under long-day conditions (Figure 11A). Whereas *otu5-1*, *hub1-4*, and *hub1-5* single mutants showed similar early flowering (~7 leaves at bolting) in comparison with Col-0 (~9.5 leaves at bolting), both double mutants flowered even earlier (~5 leaves at bolting) than the single mutants. Interestingly, as examined by RT-qPCR, the flowering times of the single and double mutants correlated with the *FLC* expression levels (Figure 11A,B). Whereas *FLC* expression levels in *hub1-4* and *hub1-5* were generally similar to those of *otu5-1* and drastically reduced in comparison to Col-0 (20–40%), further reduced *FLC* levels (≤10% of Col-0) were observed in the double mutants (Figure 11B).

### 2.12. OTU5 Is Partially Required for FLC-Mediated Flowering Suppression in Autonomous Mutants and Col-0 Expressing FRIGIDA

Notably, the delayed flowering in *FRIGIDA* (At4g00650)-containing Col-0 can be partially suppressed by crossing in *otu5-1* (Figure 11C, *FRIGIDA* vs. *FRIGIDA otu5*), suggesting that *OTU5* is partially required for *FRIGIDA*-mediated flowering suppression. Similarly, the delayed flowering of mutants harboring defects in various autonomous pathway components, such as *ld-1*, *fve-4*, and *fld-6*, can also be partially suppressed by crossing into the *otu5-1* background (Figure 11C, *ld-1*, *fve-4*, and *fld-6* vs. *ld-1 otu5*, *fve-4 otu5*, and *fld-6 otu5*, respectively), indicating that OTU5 is also partially required for autonomous pathway mutation-mediated flower suppression. Both *FRIGIDA*- and autonomous pathway mutation-mediated flowering suppression are modulated through transcriptional activation of *FLC*, which was apparently partially suppressed by crossing the respective mutants with *otu5-1* (Figure 11D). Together, these results show that *OTU5* is a novel flowering regulatory component partially required for *FRIGIDA*- and autonomous pathway mutation-mediated *FLC* activation and flowering suppression.

## 3. Discussion

Based on their distinct phylogeny, biochemical properties [9], and preliminary phenotypic analyses of T-DNA insertion mutants, the 12 identified *A. thaliana* OTU-DUBs are likely involved in various functions that are either evolutionarily conserved or specific to plants. In particular, T-DNA-inserted *otu5* null mutant plants displayed various vegetative and reproductive growth phenotypes, including early flowering. Gene ontology enrichment analysis of DEGs supported that downregulation of the major flowering suppressors *FLC* and *MAF4-5* and upregulation of *FT* and *AGL20*/*SOC1* are responsible for the early flowering phenotype of *otu5-1*. The upregulation of the flowering initiators *FT* and *AGL20*/*SOC1*, however, appears to be indirect since these genes are targets of the flowering repressors FLC and MAF4-5 [60]. Several observations support the notion that OTU5 is involved in the activation of the major flowering suppressors *FLC* and *MAF4-5* by altering the levels of various histone marks on these loci. Conspicuously, we observed reduced deposition of the activating histone mark H3K36me3 and accumulation of the repressive histone marks H3K27me3 and H3K9me2 on the *FLC* and *MAF4-5* loci in the *otu5-1* mutant (Figure 4 and Figure 5). Moreover, the OTU5 protein was found to be enriched in the nucleus, associated with chromatin (Figure 6), and even directly associated with the *FLC* and *MAF4-5* loci (Figure 7).

Taken together, our data suggest that OTU5 is a novel component involved in the epigenetic activation of major flowering repressors that modulate flowering time. Uniquely, OTU5-mediated activation of *FLC* clade loci and flowering suppression appear to be independent of the HUB1-mediated pathway. Whereas in *otu5-1* plants, ubH2B levels were slightly increased on *FLC* and *MAF4-5* loci, genome-wide, the levels of this histone mark remained unaltered in this mutant (Figure 4 and Figure 5). In contrast, homozygous *hub1* and *ubc1 ubc2* mutants showed decreased ubH2B levels genome-wide and on *FLC* clade loci [58,59]. In addition, synergistic effects on flowering and *FLC* expression were observed in the two *hub1 otu5* double mutants (Figure 11A,B). Although *otu5-1* and SWR1c subunit mutants show similar phenotypes, OTU5-mediated flowering time regulation is also independent of the SWR1c-mediated pathway. Unlike the reduced H2A.Z levels on the *FLC* and *MAF4-5* loci in *arp6-1*, the H2A.Z levels on these loci were unaltered in *otu5-1* (Figure 10). Moreover, synergistic effects on various vegetative and reproductive growth phenotypes were observed in the *otu5 arp6* double mutant compared with the single mutants. However, OTU5 is partially required for *FLC*-mediated flowering suppression in autonomous pathway mutants and *FRIGIDA*-Col as *FLC* expression, and late flowering was partially suppressed when each of these genotypes was introduced into the *otu5-1* background (Figure 11C,D). Since additional vegetative and reproductive growth phenotypes and altered transcriptomes were associated with *otu5-1*, it can be speculated that OTU5 employs similar mechanisms to regulate a further subset of genes involved in other plant functions.

### 3.1. Arabidopsis OTU DUBs OTU1, OTLD1, and OTU5 Are Involved in Epigenetic Regulation

DUBs involved in epigenetic regulation often belong to the UBP/USP class. For example, yeast (*S. cerevisiae*) Ubp8 and the corresponding human orthologue USP22 are associated with Spt-Ada-Gcn5-acetyltransferase (SAGA) coactivator complexes to modulate the histone ubiquitination/deubiquitination cycle during transcriptional activation, elongation, splicing, and mRNA export of multiple genes [61,62]. SAGA removes ubiquitin from mono-ubiquitinated histone H2B (ubH2B) to promote transcriptional elongation by facilitating Ctk1 recruitment and phosphorylation of the C-terminal RNA polymerase II [3]. In contrast, yeast Ubp10 of the Sir complex deubiquitylates ubH2B and is involved in telomere and rDNA silencing [63,64]. Ubp10 deletion resulted in accumulation of ubH2B and methylation of K4 and K79 of histone H3 (H3K4 and H3K79) to prevent association of the Sir proteins with silent loci. Moreover, Ubp10 overexpression caused a reduction of ubH2B and methylated H3K4 and H3K79 levels, allowing for the spread of the Sir complexes to non-silenced regions [63,64]. Similarly, fruit fly (*Drosophila melanogaster*) USP7 is associated with silenced genomic regions, such as telomeric domains, and is suggested to contribute to Polycomb-mediated silencing through ubH2B deubiquitylation [65].

Some Arabidopsis UBP DUBs have been shown to be involved in epigenetic regulation. Working together with the PRC1-like complex to deubiquitinate ubH2A and prevent loss of H3K27 tri-methylation, UBP12 and UBP13 are involved in the repression of specific Polycomb group (PcG) target genes such as *CONSTANS* and *FLOWERING LOCUS T*, which are critical for circadian clock and photoperiodic flowering regulation [66,67,68]. Similar to yeast Ubp8 and human USP22, Arabidopsis orthologue UBP22 is likely the catalytic subunit of the DUB module and plays a role in the H2B ubiquitylation/deubiquitylation cycle during SAGA-mediated transcriptional elongation [69,70]. Interestingly, UBP26 deubiquitylates ubH2B and is involved in histone modification to modulate transcription. UBP26 is required for the repression of the PcG complex-targeted gene *PHERES1* through H3K27 trimethylation [71]. In contrast, UBP26 appears to be also required for *FLC* activation to suppress flowering [45]. In the *ubp26* mutant, ubH2B accumulated globally and at the *FLC* locus. Moreover, H3K36me3 and H3K27me3 levels were reduced and increased, respectively, at the *FLC* locus to suppress transcription.

As described above (see Section 1), thus far, only a single human OTU DUB, TRABID/ZRANB1, has been implicated in indirect epigenetic regulation. However, all plant OTU DUBs, including Arabidopsis OTU1, OTU5, OTLD1, and rice OTLD1 orthologue (OsOTU2), that have been investigated so far for in vivo functions belong to cross-kingdom conserved clades and play a role in epigenetic regulation.

Arabidopsis OTU5 does not appear to be involved in the global modification of major histone marks because the overall chromatin levels of H3K4me3, H3K36me3, H3K36me1, H3K27me3, and ubH2B were not altered in *otu5-1* plants in comparison with the wild type (Figure 4A). However, qChIP results indicate that reductions in the activating histone mark H3K36me3 and increases in the repressive histone marks H3K27me3 and H3K9me2 were detected in various regions of the *FLC* and *MAF4-5* loci in *otu5-1* mutant plants. Moreover, a slight ubH2B increment was also detected in various regions of *FLC* and *MAF4-5* loci in *otu5-1* plants. Our qChIP results clearly suggest that *A. thaliana* OTU5 is involved in modulating the major histone marks on specific genes, such as *FLC* and *MAF4-5*. The average qCHIP yields for activating mark H3K36me3 and repressive mark H3K27me3 were found to be higher for *FLC* than for *MAF4* and *MAF5* in both wild-type and *otu5-1* plants. The magnitudes of the decrease or increase in *otu5-1* in comparison with the wild type for H3K36me3 and H3K27me3 were found to be larger in various regions of *FLC* than for *MAF4* and *MAF5* (Figure 4 and Figure 5). Our results indicate that H3K36me3 and H3K27me3 are major activating and repressive histone marks on *FLC* clade loci and are in agreement with a stronger suppression effect on the expression of *FLC* than *MAF4* and *MAF5* in *otu5-1* (Figure 3B).

### 3.2. OTU5 Isoforms Are Enriched in the Nucleus, Associated with Chromatin, and Found on FLC and MAF4-5 Loci

To directly or indirectly modify histone marks on specific genes, OTU5 needs to be localized to the nucleus. Nuclear enrichment of both OTU5 isoforms was observed when examined by subcellular fractionation. An adventitious observation of nucleus-enriched RPT2 implies a nucleus-associated function. Interestingly, the role of RPT2a in DNA methylation was previously observed [72]. Enhanced nuclear fluorescence of N- and C-terminally attached GFP fusions of both OTU5 isoforms corroborates the nuclear localization of OTU5 (Figure 6C). Furthermore, both OTU5 isoforms are capable of associating with chromatin, supported by the disruption of OTU5 precipitation with chromatin when the nuclei preparation was pretreated with MNase (Figure 6D). Nucleus- and chromatin-associated OTU5 could potentially be a critical mechanistic component involved in post-translational histone modification of specific target genes, such as *FLC* and *MAF4-5*. The direct association of OTU5 with *FLC* and *MAF4-5* was supported by qChIP experiments (Figure 7).

### 3.3. OTU5 Acts Independently of SWR1c-Mediated FLC-Activation and Flowering Suppression

It is currently unknown how OTU5 or the OTU5-associated complex(es) modulate the levels of various histone marks and ubH2B on *FLC* and *MAF4-5*. Although catalytic activity of OTU5 was not detected in vitro, the in vivo functional importance of the OTU5 catalytic site was supported by the observation of complementation failure with *otu5-1* and dominant negative effects in wild-type plants by the expression of a catalytic-site mutated OTU5 (Figure 1 and Figure 2). It is possible that the deubiquitination activity of Arabidopsis OTU5 can only be activated when the protein is post-translationally modified, for instance, by phosphorylation [18,73] or when associated with interacting proteins or complexes, as exemplified by and proposed for other characterized DUBs [57].

The dominant negative effects observed when overexpressing the catalytic site-mutated OTU5 in the Col-0 background suggest that the wild-type OTU5 could potentially exert function in association with yet uncharacterized proteins or complexes. OTU5-associated complex(es) were indeed observed by size-exclusion chromatography (Figure 8). Although a large portion of the OTU5 isoforms associated with the low-molecular-weight fractions, a substantial quantity of OTU5 isoforms were observed in fractions peaking at ~500 kDa, a size similar to that of SWR1c [48]. Interestingly, *OTU5*-null mutant plants generally displayed phenotypes similar to those of mutants that are defective in SWR1c subunits and histone H2A.Z, indicating that OTU5 could potentially be a novel component of SWR1c. However, the available evidence suggests that the OTU5-associated protein complex(es) is distinct from SWR1c and OTU5 likely functions independently of the SWR1c-mediated pathway. First, the isolated yeast SWR1c contains 14 subunits, and none of these subunits is an OTU5 homolog or DUB [46,74,75]. Second, the molecular weights of OTU5-associated protein complexes were essentially unaltered in the absence of ARP6, a subunit critical for forming an SWR1c subcomplex [74,75]. Third, distinct phenotypes not observed in *otu5-1* plants, such as extra floral organs and serrated rosette leaves, were reported in mutants defective in genes encoding SWR1c subunits and H2A.Z [47,51,52,53,54]. Fourth, unlike *arp6-1* [41], de-repression of phosphate starvation response genes was not observed in *otu5-1* plants grown under phosphate-replete conditions [40]. Fifth, in agreement with the requirement of ARP6 for H2A.Z deposition on *FLC*, expression of *FLC* and *MAF4-5*, and flowering repression [48,51], we observed a reduction in H2A.Z levels on *FLC* and *MAF4-5* in *arp6-1*. However, alteration of H2A.Z levels on the *FLC* and *MAF4-5* loci was not observed in *otu5-1* plants (Figure 10). Sixth, whereas FRI-mediated and two autonomous pathway mutations (*fve-1*- and *fca-1*)-mediated late flowering and *FLC* activation were nearly completely abolished in the *arp6-1* mutant background [51,76], late flowering and *FLC* expression were only partially suppressed in *FRI-Col-0*, and three tested autonomous pathway mutations, including *fve-1* in the *otu5-1* background (Figure 11).

Furthermore, pronounced synergetic phenotypes of both vegetative and reproductive growth were observed in the *otu5-1 arp6-1* double-mutant plants in comparison with each of the single mutants (Figure 9). For vegetative growth phenotypes, the double mutant displayed synergetic effects on seedling growth rate, rosette leaf number and size, final plant height, primary root length, and root hair length/density. Although both the double- and single-mutant plants showed smaller flowers in comparison with Col-0, dramatic synergetic fertility defects were associated with double-mutant plants, including the shortest pistil and silique length, smallest ovule number, and highest ovule abortion rate. However, although the lowest expression level of *FLC* was observed in *otu5-1 arp6-1* double-mutant plants, the early flowering time was not further advanced, suggesting that the threshold of *FLC* in suppressing flowering may have been surpassed in the double-mutant plants. Alternatively, flowering time is likely regulated by the total expression levels of all *FLC*-related flowering suppressors or by the concerted effects from different flowering pathways.

Moreover, based on the identified DEGs from the transcriptome analyses of the genotypes under investigation, ARP6 appears to play a more general role and exerts a stronger effect on overall transcriptome changes than OTU5. In support of this supposition, the analysis of congruent DEGs indicated that a large portion of *otu5 arp6* DEGs were contributed by a loss of ARP6. A previous transcriptome analysis also suggested a pronounced effect on overall transcriptome change (622 DEGs detected) associated with mutant defective in a SWR1c subunit (*pie1-5*) [55]. Moreover, a significant portion of *otu5 arp6* DEGs that overlapped with those in single mutants showed synergetic expression effects in comparison with each of the single mutants (Appendix A). Of note, *FLC* transcripts were further reduced in *otu5 arp6* relative to single mutants. Overall, our results support the distinct mechanisms likely associated with OTU5- and ARP6-mediated functions.

Although OTU5- and SWR1-associated protein complex(es) are likely distinct entities, evidence suggests a possible functional overlap between these complexes. First, the similar phenotypes of *OTU5*-null plants and mutants defective in genes encoding SWR1c subunits and H2A.Z suggest that OTU5- and SWR1 complex(es) could act in parallel pathways. Analysis of overlapping DEGs suggests that ARP6 likely covers the majority of OTU5-associated functions (Appendix A). Analyses of expression patterns in other genotypes for DEGs detected in *otu5*, *arp6*, and the double mutant also suggest that ARP6 likely executes substantial OTU5-associated functions (Appendix A). Moreover, although the size of the OTU5-associated complex(es) did not shift in the absence of ARP6, the abundance was clearly reduced. This suggests that the assembly, stability, and/or regulation of the OTU5-associated complex(es) was affected by the absence of ARP6.

### 3.4. OTU5 Acts Independently of HUB1- and UBP26-Mediated FLC-Activation and Flowering Suppression

The evolutionary conserved HUB1-2 and UBC1-2 catalyze the mono-ubiquitination of H2B on a C-terminal conserved lysine [58]. Based on the synergistic effects observed on *FLC* expression and flowering time when *otu5-1* was crossed into two *hub1* alleles (Figure 11A,B), the OTU5-mediated activation of *FLC*-related flowering repressors and flowering suppression are likely also independent of the HUB1-2/UBC1-2-mediated flowering suppression pathway. The synergistic effects observed are well perceived, as OTU5 and HUB1 are likely to carry out opposite catalytic activities. Furthermore, additional distinct phenotypes, such as reduced seed dormancy, pale green leaves, and a bushy appearance, were associated with *hub1* alleles but not *otu5-1* [58,77]. HUB1-2/UBC1-2-mediated ubiquitination plays a major role in tuning ubH2B levels genome-wide and specifically on *FLC* clade loci to modulate levels of the activating histone marks H3K4me3 and H3K36me2 but not that of H3K9me2 [58]. In contrast, in the *otu5-1* mutant, genome-wide ubH2B levels remained unchanged, and slightly increased levels of H3K9me2 and ubH2B were associated with the *FLC*, *MAF4*, and *MAF5* loci. Moreover, while mutation of HUB1-2/UBC1-2 components had more pronounced effects on the expression of *MAF4* and *MAF5* than that of *FLC* [58], loss of OTU5 appears to be more critical for the expression of *FLC* than that of *MAF4* and *MAF5*.

OTU5-mediated flowering suppression also appears to be independent of the UBP26-mediated pathway. Genome-wide increases in ubH2B and H3K4me3 deposition were observed in *ubp26-1* [45,78], but not in *otu5-1*, and the effects of the loss of UBP26 on the expression of various members of the *FLC* clade are different from those observed in plants without functional OTU5. In *ubp26-1* plants, strongly reduced expression was detected for *FLC*, followed by *MAF2* and *MAF3*, while *MAF1* and *MAF4* transcripts were only slightly reduced, and the expression of MAF5 even increased [45].

However, the novel OTU5-mediated pathway described here appears to be partially required for *FLC*-mediated flowering suppression in autonomous pathway mutants and *FRIGIDA*-Col. To determine how OTU5 or the OTU5-associated complex(es) modulate the levels of various histone marks and ubH2B on *FLC* and *MAF4-5*, it is essential to isolate the in vivo OTU5-interacting proteins or OTU5-associated complex(es) and determine the identities of their constituents and in vivo substrates. It would be interesting to test whether OTU5 uses ubH2B as a substrate. Additionally, the pleiotropic phenotypes of *OTU5*-null mutant plants suggest that OTU5 could be involved in more cellular processes by targeting and regulating a subset of target genes. It can be speculated that OTU5 employs mechanisms similar to those observed for *FLC* and *MAF4-5* regulation. Further, it would be interesting to screen for more OTU5 target genes and their respective cellular functions. It would be equally intriguing to examine whether the potential OTU5 orthologues, human OTUD6A/OTUD6B, yeast Otu2, and rice (*O. sativa*) orthologous products (04g0619500) employ similar mechanisms to regulate specific processes.

## 4. Materials and Methods

### 4.1. Plant Materials and Growth Conditions

*OTU* T-DNA-inserted mutants (Appendix A), *ld-1* [79], two *hub1* mutant alleles (*hub1-4*/Salk_122512 and *hub1-5*/Salk_044415) [58], and FRIGIDA-Col [80] were requested from the Arabidopsis Biological Resource Center at Ohio State University (Columbus, OH, USA) or the European Arabidopsis Stock Center at the University of Nottingham (Loughborough, UK). The *arp6-1* (SAIL_599_G03) [51], *fld-6* (SAIL_642_C05.V2) [81], and *fve-4* [82] mutants were described previously and are kind gifts from Yee-Yung Charng (ABRC, Academia Sinica, Taiwan) and Keqiang Wu (National Taiwan University, Taiwan). To grow *A. thaliana* plants, seeds were surface sterilized and stratified in distilled water at 4 °C for three days in the dark. The stratified seeds were then sown in soil (a 6:1:1 mixture of humus:vermiculite:perlite) and grown in a 16-h light/8-h dark or 8-h light/16-h dark photoperiod with a light intensity of ~120 µmol·m^−2^·s^−1^ at 22 °C. To select transgenic plants, sterilized seeds were stratified on half-strength Murashige and Skoog plates (0.8% agar, pH 5.8) supplemented with 1% sucrose and appropriate antibiotics at 4 °C for three days in the dark. The seeds were then germinated in a growth chamber at 22 °C (16-h light/8-h dark). Twelve-day-old seedlings were transferred to soil and grown under the same light/dark photoperiods and light intensity described above. To determine primary root lengths, stratified seeds were germinated and grown on half-strength Murashige and Skoog plates (1.2% agar and 1.5% sucrose) in a growth chamber at 22 °C (16-h light/8-h dark) for five days; seedlings of similar sizes were then transferred to new plates and grown vertically under the same conditions for an additional seven days. To determine the root hair length and density, five-day-old seedlings were transferred to phosphate-replete conditions, as previously described [41] for an additional two days.

### 4.2. RT-PCR and RT-qPCR

Total RNA was isolated from 14-day-old seedlings with Trizol (Life Technologies, Carlsbad, CA, USA), treated with DNase using the TURBO DNA-free kit (Thermo Fisher Scientific, Waltham, MA, USA) according to the manufacturers’ instructions, and quantified by NanoDrop ND-1000 (Thermo Fisher Scientific). A total RNA of 2 µg was used for first-strand cDNA synthesis in a 20 µL reaction with the Transcriptor First Strand cDNA Synthesis Kit (Roche Life Science, Indianapolis, IN, USA). Sterile water was added to a final volume of 150 µL. To check for the presence of corresponding full-length transcripts (ORFs) for various T-DNA-inserted mutants, 40 extended amplification cycles were conducted. *UBQ10* was examined as a loading control. The primer pairs for the full-length transcripts of the *OTU* loci were the same as those used for making recombinant constructs, as previously described [9]. RT-qPCR was performed in triplicate with first strand cDNA samples and gene-specific primers on a QuantStudio 12K Flex Real-Time PCR System (Applied Biosystems, Austin, TX, USA). Each of the reaction mixes comprised 2X SYBR Green PCR Master Mix (Applied Biosystems), forward and reverse primers, first strand-cDNA template, and sterile water. The reactions were added to 96- or 384-well clear reaction plates (Applied Biosystems), sealed with an optical adhesive cover (Applied Biosystems), and placed into a real-time PCR machine. The reactions were incubated at 50 °C for 2 min, denatured at 95 °C for 10 min, followed by 40 cycles of denaturation at 95 °C for 15 s and extension at 60 °C for 1 min. *UBQ10* was also amplified in triplicate and used as an internal control. The specificity of each pair of gene-specific primers was determined using a dissociation curve. The relative expression level of each gene was calculated by the difference between the cycle threshold (Ct) of target genes and *UBQ10* (ΔCt = Ct^target gene^ − Ct*^UBQ10^*), that is 2^−ΔCt^ [83]. The primer pairs for the UBQ10 control and flowering regulatory genes are listed in Appendix A.

### 4.3. Cryo-SEM and Light Microscopy

For microtome sectioning, inflorescence stems were cut just below the first internode and fixed in 4% glutaraldehyde for 2 h, followed by embedding in Leica Historesin according to the manufacturer’s instructions. Sections of 10 µm were made on a rotary microtome, and the tissues were stained with 0.5% toluidine blue. The leaf areas were measured using ImageJ 1.38× (National Institutes of Health, Bethesda, MD, USA). To determine the cell densities of the leaves and petals, cryo-SEM was conducted, as previously described [84].

### 4.4. Transgenic Constructs and A. thaliana Transformation

To generate complementation or overexpression lines with *otu5-1* or Col-0, 35S promoter and coding sequence of either wild-type OTU5a or OTU5a-C192S from entry vectors (pENL4-2-L3, pENTR221-OTU5a, and pENTR221-OTU5a-C192S, respectively) were mobilized together into gateway vector pKNGSTAP [85] using Gateway Technology Clonase II according to the manufacturer’s instruction (Invitrogen). The wild-type and C192S-mutated OTU5a coding sequences were first PCR-amplified using PfuTurbo (Agilent Technologies, Santa Clara, CA, USA) from pET28a-OTU5a and pET28a-OTU5a-C192S, respectively, with the primer pair OTU5a_CF and OTU5a_CR (Appendix A) and cloned separately into pDONR221 (Life Technologies) to give pENTR221-OTU5a and pENTR221-OTU5a-C192S. The entry vector pENL4-2-L3 was requested from VIB, Belgium. The vector pET28a-OTU5a-C192S was derived by site-directed mutagenesis in accordance with the manufacturer’s instructions (Agilent Technologies) from pET28a-OTU5a, described previously [9] using the primer pair OTU5a-C192S-T and OTU5a-C192S-B (Appendix A). A freeze–thaw method was used to transform *Agrobacterium tumefaciens* GV3101, and the *A. thaliana* transformation was performed as previously described [86]. Homozygous complementation and overexpression lines were selected from the T3 plants.

### 4.5. Transcriptome Analyses

For the first *otu5-1* transcriptome comparison with Col-0, antisense-amplified RNA (aRNA) containing aminoallyl-UTP was prepared from 1 µg DNase-treated total RNA from 10-day-old seedlings, collected in the morning, using the Amino Allyl MessageAmp II aRNA amplification kit (Applied Biosystems/Ambion). The quantity and quality of total RNA were checked with an Agilent 2100 bioanalyzer using an RNA 6000 nano assay kit (Agilent Technologies). The quantity and quality of modified aRNA were checked with an Agilent 2100 bioanalyzer and a NanoDrop ND-1000 UV–Vis spectrophotometer (Thermo Fisher Scientific). The Col-0 and *otu5-1* aRNAs were subsequently labeled with Alexa Fluor 555 and Alexa Fluor 647 reactive dyes (Life Technologies), respectively, purified by the Qiagen RNeasy MiniElute Cleanup kit (Qiagen, Hilden, Germany), and mixed in equal amounts (825 ng each). An *A. thaliana* genome 44K oligo microarray kit (G2519F-021169, Agilent Technologies) was used for hybridization. Microarray hybridization and washing were performed with the Agilent Gene Expression Hybridization kit and Gene Expression wash pack according to the manufacturer’s instructions. The chip was scanned with an Agilent G2565CA microarray scanner. The array image was acquired with Feature Extraction software version 10.7.1.1 using the GE2-107_Sep09 protocol, and the array data were analyzed using GeneSpring GX 11.2 (Agilent Technologies). Differentially expressed entries with fold changes of ≥2 and *p*-values of <0.05 were collected. Gene Ontology enrichment was analyzed using GOBU’s MultiView plugin (https://gobu.sourceforge.io/, accessed on 6 April 2011) based on the TAIR GO annotations of all protein-coding genes of TAIR10 genome release [43,44]. The enriched GO terms with *p*-values of <0.01 derived by Fisher’s exact test [87] and TopGo’s *elim* algorithm [88] were collected.

For transcriptome comparison of *otu5-1*, *arp6-1*, and *otu5-1 arp6-1* with Col-0, samples used for total RNA isolation were collected in the evening from 10-day-old seedlings grown on 1/2 MS medium. Total RNA was purified using the RNeasy Mini Kit (Qiagen). Total RNA quality (OD260/280 > 2.0; RNA integrity number > 7.0) was analyzed by ND-1000 spectrophotometer (Thermo Fisher Scientific) and Bioanalyzer expert RNA 6000 Nanochip (Agilent Technologies). Labeling and hybridization of cRNA were performed using the One-Color Microarray-Based Gene Expression Analysis system (Agilent Technologies) according to the manufacturer’s protocol. Labeled cRNA was synthesized and amplified from 0.2 μg total RNA using a Low Input Quick-Amp Labeling kit (Agilent Technologies) and Cy3 (CyDye, Agilent Technologies). Cy3-labled cRNA of 1.65 μg was fragmented at 60 °C for 30 min. The fragmented cRNA was then pooled and hybridized to Arabidopsis (V4) 4×44K Microarray (Agilent Technologies) at 65 °C for 17 h. After the washing and drying steps, the microarrays were scanned with an Agilent microarray scanner (Agilent Technologies) at 535 nm for Cy3. Array intensity files were processed using the Bioconductor package *limma* [89], where normalization was performed using its normalizeVSN function [90] and a differential expression analysis was performed using its lmFit and eBayes functions [91]. Differentially expressed entries with fold changes of ≥2 and *p*-values of <0.05 were collected. Venn diagrams were made using Venneuler.

### 4.6. Immunoblotting and Monitoring Global Levels of Various Histone Marks

Extraction of total proteins and immunoblotting were conducted as previously described [92]. Rabbit polyclonal antibodies were raised against purified recombinant full-length *A. thaliana* OTU1, OTU2, OTU5b, RPN8a, RPN10, RPN13, RAD23c, DDI1, and NUB1 (custom-made by Genesis Biotech or Cashmere Biotech, Taipei, Taiwan). Rabbit polyclonal antibodies made against *A. thaliana* PBA1, RPT2, RPT5, RPN5, RPN6, and RPN12 were purchased from Enzo Life Sciences. The global levels of H3 marks and ubH2B in wild-type and *otu5-1* plants were detected from nuclear-enriched protein extracts of 10-day-old seedlings following the chromatin extraction and sonication of the ChIP protocol, and the chromatin solution was diluted with 2× loading buffer and denatured at 100 °C for 10 min. The samples were examined by immunoblotting with specific rabbit polyclonal antibodies against H3K36me3 (ab9050, Abcam), H3K27me3 (ABE44, Millipore), H3K36me1 (ab9048, Abcam), histone H3 (ab1791, Abcam), or mouse monoclonal antibodies against H3K4me3 (ab12209, Abcam) or ubH2B (MM-0029, Médimabs). A chemiluminescent system (Perkin Elmer) was used to develop the protein blots according to the manufacturer’s instructions.

### 4.7. Quantitative Chromatin Immunoprecipitation (qChIP)

The ChIP experiment was performed as described in [93,94]. Fourteen-day-old seedlings were harvested and cross-linked with 1% formaldehyde under a vacuum for 15–20 min. The cross-linking reaction was stopped with 125 mM glycine, followed by incubation for 10 min with occasional shaking. Chromatin was extracted and sheared to an average length of 500–1000 bp through sonication by Biorupter (Diagenode), and then sheared chromatin was preincubated at 4 °C for 1 h with pre-equilibrated protein-A agarose beads (Millipore), followed by centrifugation at 4000 rpm for 2 min. The resulting supernatant was incubated at 4 °C overnight with specific rabbit polyclonal antibodies against H3K36me3 (ab9050, Abcam), H3K27me3 (ABE44, Millipore), H3K9me2 (07-441, Millipore), H2A.Z (H2A.Z isoforms HTA9 and HTA11), OTU5b (see above), or mouse monoclonal antibodies against the ubH2B (MM-0029, Médimabs), followed by the addition of protein A agarose beads (40 μL) and further incubation for 3–4 h. The α-H2A.Z was custom-made (LTK BioLaboratories, Taipei, Taiwan) and a kind gift from Yee-Yung Charng (ABRC, Academia Sinica, Taiwan). The immunocomplexes attached to the protein A agarose beads were sequentially washed with low- and high-salt wash buffers (20 mM tris-HCl/pH 8.0, 2 mM EDTA, 0.1% SDS, 1.0% Triton X-100) supplemented with 0.15 M NaCl (low salt) or 0.5 M NaCl (high salt), then with a LiCl wash buffer (250 mM LiCl, 10 mM tris-HCl/pH 8.0, 1.0 mM EDTA, 1% NP-40, 1.0% deoxycholate), and twice with a TE buffer (10 mM tris-HCl/pH 8.0, 1.0 mM EDTA), and eluted from the beads by incubation at room temperature for 15 min in elution buffer (0.1 M NaHCO_3_, 0.5% SDS). The cross-links were heat reversed at 65 °C, and the DNA was purified by a QIAGEN QIAquick gel extraction kit. The precipitated DNA samples associated with the modified histone H3 were quantified with real-time PCR using SYBR Green PCR Master Mix through QuantStudio 12K Flex Real-Time PCR System (Applied Biosystems) with programs recommended by the manufacturer. The comparative threshold cycle (CT) method was used to determine the relative qCHIP yields [95]. The percent input for histone modifications was calculated using the total input DNA as a control in all experiments. The sequences of the PCR primer pairs are listed in Appendix A. All qChIP results presented are the averages of the three biological replicates.

### 4.8. Isolation of Total Proteins, Cytosolic Proteins, Nuclear Extracts, and Chromatin

Cytosolic, nuclear, and chromatin fractions were isolated as previously described [96,97]. Tissues from 15 DAS plants were homogenized in buffer A (2.5% Ficoll 400, 5% dextran T40, 0.4 M sucrose, 25 mM Tris-HCl, pH 7.4, 10 mM MgCl_2_, 10 mM β-mercaptoethanol, and 1 mM DTT) supplemented with a 1x protease inhibitor cocktail (Roche) using a mortar and pestle and then filtered through two layers of Miracloth (EMD Millipore). Triton X-100 was added to a final concentration of 0.5%, and the mixture was incubated on ice for 15 min. The nuclei pellet (P1) was collected by low-speed centrifugation (1300× *g*) for 4 min at 4 °C. The supernatant (S1) was further clarified by high-speed centrifugation (20,000× *g*) for 15 min at 4 °C to remove cell debris and insoluble aggregates as the cytosolic fraction (S2). The nuclei pellet (P1) was washed with buffer A containing 0.1% Triton X-100, resuspended gently in 1 mL of buffer A, and transferred to a microcentrifuge tube. This nuclei-enriched preparation was centrifuged at 100× *g* for 1 min to pellet the starch and cell debris. The supernatant was subsequently centrifuged at 1800× *g* for 5 min to pellet the nuclei (P2), which were washed once in buffer A and then lysed in buffer B (3 mM EDTA, 0.2 mM EGTA, and 1 mM DTT) supplemented with 1x protease inhibitors (as nuclear fraction). Insoluble chromatin was collected in pellet form by centrifugation (1700× *g*) for 4 min at 4 °C, washed once in buffer B, and centrifuged again under the same conditions. The supernatant was saved as the nucleoplasmic fraction (S3). The final chromatin pellet (P3) was resuspended in a 2× SDS-PAGE sample buffer and sonicated for 15 s with an XL2020 microtip (Misonix) at 25% amplitude. To release chromatin-bound proteins by nuclease treatment, nuclei (P1) were resuspended in buffer A supplemented with 1 mM CaCl_2_ and 0.2 U of micrococcal nuclease (New England Biolabs, Ipswich, MA, USA). After incubation at 37 °C for 1 min, the nuclease reaction was stopped by the addition of 1 mM EGTA. Nuclei and chromatin isolation were conducted as described above. Total protein extracts were prepared from tissues by homogenization in liquid nitrogen. Fifty milligram samples were resuspended in a 2× SDS-PAGE sample buffer and boiled for 5 min, and the cell debris was removed by centrifugation before loading onto 10% SDS-PAGE gels.

### 4.9. Transient Expression in A. thaliana Protoplasts

*GFP* fusion constructs of *OTU5a* and *OTU5b* for transient expression experiments were constructed in p2X35S-TEV-eGFP-35ST (Figure 6B) derived from a T & A cloning vector (Real Biotech Co., Taipei, Taiwan). The CaMV double 35S promoter (2× P_35S_-TEV) and CaMV 35S poly(A) addition signal (TER_35S_) were PCR derived from pSAT5-DEST-EYFP-C1 [98], and the eGFP coding sequence was PCR derived from pZm13::GFP [99], with modifications at the 5′ and 3′ ends of five GA repeat coding sequences. For N-terminal fusion to eGFP, the OTU5a and OTU5b coding sequences were amplified and cloned in frame to 5′ of the eGFP coding region using the primer pairs OTU5a-X/OTU5-E and OTU5b-X/OTU5-E, respectively (Appendix A). The 5′ and 3′ primers were designed to add *Xma*I and *Eco*RI restriction sites, respectively, for cloning. For C-terminal fusion to eGFP, the OTU5a and OTU5b coding sequences were amplified and cloned in frame to 3′ of the eGFP coding region using the primer pairs OTU5a-B/OTU5-S and OTU5b-B/OTU5-S, respectively (Appendix A). The 5′ and 3′ primers were designed to add *Bam*HI and *Sal*I restriction sites, respectively, for cloning.

Protoplasts were isolated from rosette leaves of four-week-old *A. thaliana* plants grown under 12-h light/12-h dark at 22 °C by slicing them into 0.5-mm sections, followed by vacuum infiltration and incubation in digestion solution (0.4 M mannitol, 20 mM MES, pH 5.7, 20 mM KCl, 10 mM CaCl_2_, 0.1% BSA, and 5 mM β-mercaptoethanol) containing 1% cellulase R-10 and 0.25% macerozyme R-10 (Yakult Pharmaceutical Industry, Tokyo, Japan) for 4 h with shaking. Cells were filtered through Miracloth, washed three times in wash solution (150 mM NaCl, 125 mM CaCl_2_, 5 mM KCl, 2 mM MES, pH 5.7, and 5 mM glucose), and finally diluted to 2 × 10^6^ cells/mL. Protoplasts (2 × 10^5^ cells) were transfected by adding 25 µg of purified plasmids in 200 µL of wash solution, followed by the addition of 220 µL of polyethylene glycol solution (40% polyethylene glycol-4000, 240 mM mannitol, and 100 mM CaCl_2_), and incubated 10 min at room temperature. The transfected cells were then pelleted by centrifugation at 100× *g* for 2 min, resuspended in 400 µL of wash solution, and incubated for 16 h under light at room temperature. GFP fluorescence was visualized using a Zeiss LSM 510 META confocal microscope equipped with Zeiss LSM Image software v4.2 (Carl Zeiss, Oberkochen, Germany).

### 4.10. Gel Filtration Chromatography

Gel filtration was performed on a HiPrep 16/60 Sephacryl S-300HR column (Sigma-Aldrich, St. Louis, MO, USA) in a buffer consisting of 50 mM Tris, pH 7.5, 200 mM NaCl_2_, 2 mM EDTA, 10 mM MgCl_2_, and 10% (*v*/*v*) glycerol. The column was calibrated with gel filtration markers (MWGF1000, Sigma-Aldrich) containing a mixture of standard proteins ranging from 66 to 669 kDa. Plant extracts were prepared by grinding three-week-old, soil-grown *A. thaliana* plants, excluding roots in liquid nitrogen with 2 volumes of gel filtration buffer supplemented with 1 mM β-mercaptoethanol, 0.1% (*v*/*v*) Nonidet P-40, 1 mM dithiothreitol (DTT), and protease inhibitors (1 mM PMSF, 15 mM *p*-nitrophenylphosphate, 60 mM β-glycerophosphate, 0.1 mM sodium vanadate, 1 mM sodium fluoride, 10 µg/mL aprotinin, 5 µg/mL pepstatin, 10 µg/mL soybean trypsin inhibitor, and 0.1 mM benzamidine). The samples were cleared by centrifugation at 12,000 rpm for 10 min and filtered through 0.45-µm filters. Chromatography was conducted at room temperature with a flow rate of 0.25 mL/min, and 1 mL fractions were collected. The proteins in each fraction were concentrated by cold 10% TCA precipitation, washed five times with ice-cold acetone, and then analyzed for the presence of OTU5 isoforms by immunoblotting using OTU5 antisera purified by NHS HP SpinTrap (GE Healthcare). Two independent biological replicates with very similar results were obtained.

## 5. Conclusions

The cross kingdom-conserved Arabidopsis OTU5 is involved in various vegetative and reproductive growth, including organ size, root growth, flowering time, and fertility. Similar to two other investigated Arabidopsis OTU DUBs, OTLD1 and OTU1, which are involved in epigenetic regulation of specific target genes, such as *DA1* and *DA2*, Arabidopsis OTU5 is involved in epigenetic activation of the major flowering repressors *FLC*, *MAF4*, and *MAF5*. As shown schematically in Figure 12, our results support the supposition that Arabidopsis OTU5 is a novel flowering repressor, acting independently of the SWR1c- and HUB1/2-UBC1/2-mediated flowering pathways but is partially required for autonomous mutant- and FRI-mediated pathways. Failure to rescue the *otu5*-1 growth phenotypes by OTU5-CS mutation and dominant negative effects observed in OTU5-CS-overexpressing wild-type plants suggest that the conserved catalytic site of OTU5 is critical in vivo. This also suggests that OTU5 exerts an in vivo function with associated proteins or complexes, which is also supported by the presence of OTU5 complexes in vivo. To further investigate the mechanisms underlying OTU5-mediated functions, it will be essential to isolate the in vivo OTU5-interacting proteins or OTU5-associated complex(es) and determine the identities of their constituents and in vivo substrates.

## Figures and Tables

**Figure 1 ijms-24-06176-f001:**
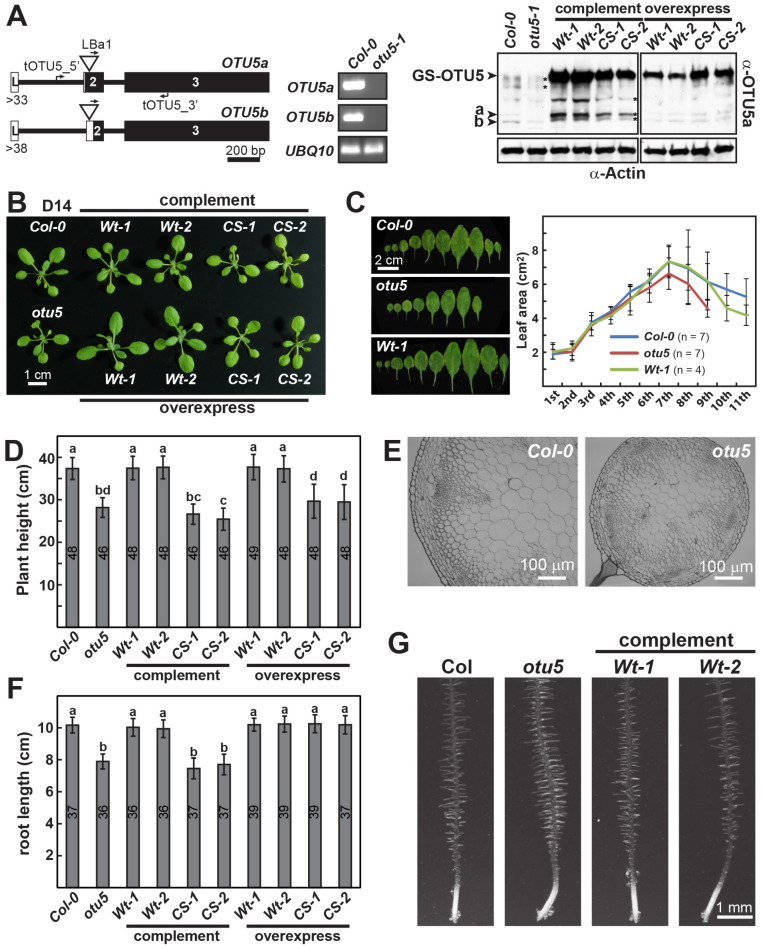
Vegetative growth phenotypes of *otu5-1* mutant plants. (**A**) Full-length *OTU5a* and *OTU5b* transcripts and their encoded proteins were not detected in *otu5-1*. Left, schematic diagrams showing the T-DNA insertion sites (triangles, not in scale) in *OTU5a* and *OTU5b*. Exons are denoted by numbers in boxes, with coding regions shaded in black; introns are indicated by lines. L, leader exon. The positions of the genotyping primers tOTU5_5′, tOTU5_3′, and T-DNA left border primer LBa1 are indicated (Appendix A). Middle, expression of the full-length *OTU5a* and *OTU5b* transcripts was examined by RT-PCR in Col-0 and *otu5-1*. The *UBQ10*-amplified fragment served as a loading control. Right, expression of OTU5 isoforms (a and b, arrowheads) in Col-0 and *otu5-1* and expression of N-terminal GS-tagged wild-type and catalytic site-mutated OTU5a (GS-OTU5, arrowhead) in complemented and overexpressing lines (*Wt-1*, *Wt-2*, *CS-1*, and *CS-2*). Nonspecific background signals are marked with asterisks. The source of these unspecific signals has not been investigated. Immunoblotting using α-OTU5b was performed with total protein extracts isolated from 15-day-old seedlings. Duplicated samples were analyzed using antisera against actin (α-Actin) to verify equal loading. (**B**) Representative seedlings 14 DAS (Col-0, *otu5-1*, and various complemented and overexpressed lines). (**C**) Reduced number and size of rosette leaves in *otu5-1* plants. A representative set of rosette leaves at 27 DAS in order of production is shown for Col-0, *otu5-1*, and a GS-OTU5a-complemented *otu5-1* line (left). Average area of each rosette leaf in order from 4–7 plants of Col-0, *otu5-1*, and a GS-OTU5a-complemented *otu5-1* line at 27 DAS (right). (**D**) Average plant heights of Col-0, *otu5-1*, and various complemented and overexpressed lines at 45 DAS (*n* = 46–49). (**E**) Representative toluidine blue-stained sections of primary inflorescence stems for Col-0 and *otu5-1* plants. (**F**) Average primary root lengths of seedlings for Col-0, *otu5-1*, and various complemented and overexpressed lines grown under long-day conditions at 12 DAS (*n* = 36–39). (**G**) Representative root tips of seedlings for *otu5-1*, Col-0, and two complemented lines at seven DAS grown under phosphate-replete conditions. Different letters denote significant differences by pairwise comparison using Student’s *t*-test; error bars indicate standard deviations.

**Figure 2 ijms-24-06176-f002:**
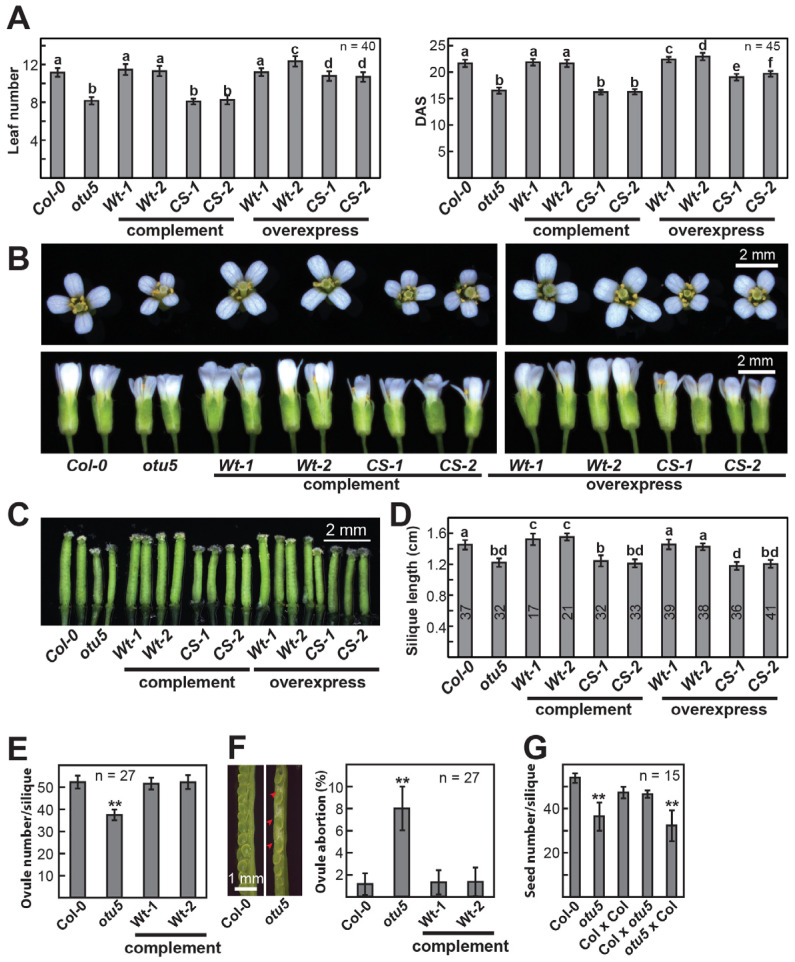
Reproductive growth phenotypes of *otu5-1* plants. (**A**) Flowering times measured either by rosette leaf number (left) or DAS (right) at bolting for Col-0 and *otu5-1* plants, and various complemented and overexpressed lines grown under long-day conditions. (**B**) Top or side view of representative flowers. (**C**) Representative gynoecia/pistils of mature flowers. (**D**) Averaged lengths of matured siliques (numbers of siliques averaged are indicated). (**E**) Averaged ovule number of each silique. (**F**) A significant proportion of aborted ovules (red arrows) was observed in the mature siliques of *otu5-1*. Left, representative dissected siliques from Col-0 and *otu5-1* plants. Right, ovule abortion rate. (**G**) Seed numbers for each silique for selfed Col-0 and *otu5-1* plants and for control (Col × Col) and reciprocal crosses between Col-0 and *otu5-1* plants. Different letters denote significant differences by pairwise comparison using Student’s *t*-test. ** *p* < 0.01, *otu5-1* was compared with Col-0, or selfed *otu5-1* plants, control, and reciprocal crossed plants were compared with selfed Col-0 plants using Student’s *t*-test. Sample sizes are indicated (*n*). Error bars indicate standard deviations.

**Figure 3 ijms-24-06176-f003:**
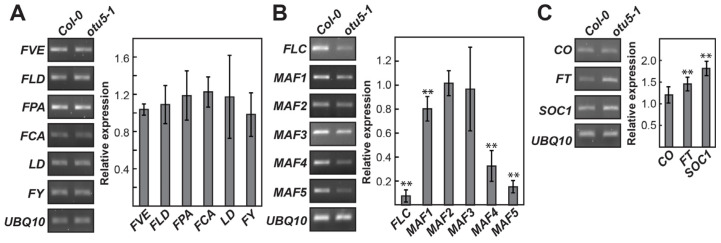
Expression of various flowering regulators and autonomous pathway components in *otu5-1* plants. Expression levels in Col-0 and *otu5-1* seedlings of 10 (RT-PCR) or 14 DAS (RT-qPCR) for major autonomous pathway components, including *FVE*, *FLOWERING LOCUS D*, *FPA*, *FCA*, *LUMINIDEPENDENS*, and *FY* (**A**), major flowering repressors *FLC* and *FLC*-related flowering repressors *MAF1-5* (**B**), and major flowering activators *CO*, *FT*, and *SOC1* (**C**) were examined by RT-PCR (left panels) and RT-qPCR (right panels). *UBQ10* was used as a control. ** *p* < 0.01, significance of expression levels for various flowering regulators in *otu5-1* were compared with those in Col-0 using Student’s *t*-test.

**Figure 4 ijms-24-06176-f004:**
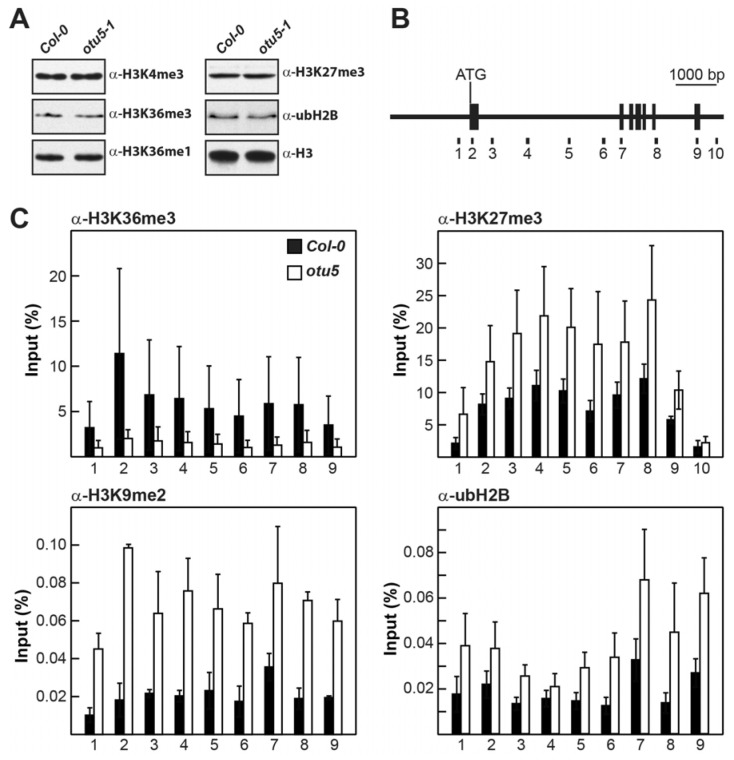
Global and *FLC*-associated levels of major activating and repressive histone marks in *otu5-1* plants. (**A**) Global levels of various histone marks, including H3K4me3, H3K36me3, H3K36me1, H3K27me3, and ubH2B, of nuclear-enriched protein extracts prepared from ten-day-old Col-0 and *otu5-1* seedlings were examined by immunoblotting using antibodies as indicated. Levels of histone 3 (H3) were examined as a loading control. (**B**) A schematic diagram in scale of the *FLC* gene structure depicts 10 amplification regions (bars) labeled 1–10 that were analyzed by qChIP. Sequences upstream of the start codon (ATG), downstream of the stop codon, and introns are represented by lines; exons are indicated by boxes. The corresponding primer pairs are listed in Appendix A. (**C**) The percentage qChIP amplification yields, in comparison with those amplified from input DNA, from 9–10 *FLC* regions using various antibodies, as indicated against H3K36me3, H3K27me3, H3K9me2, and ubH2B for *otu5-1* and Col-0 seedlings. Error bars indicate standard deviations.

**Figure 5 ijms-24-06176-f005:**
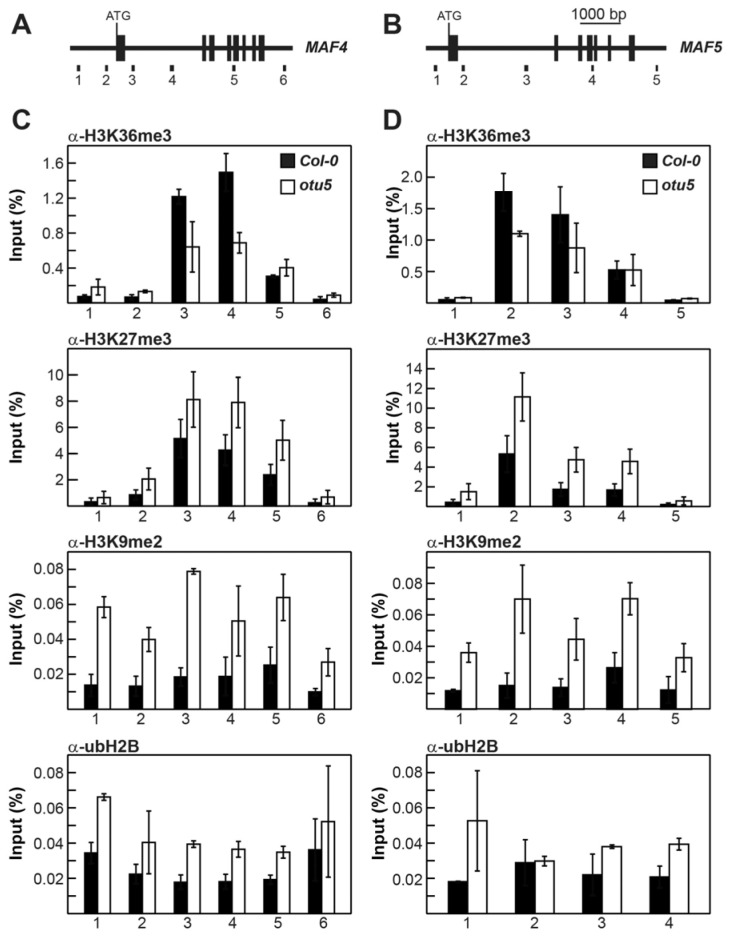
*MAF4*- and *MAF5*-associated levels of major activating and repressive histone marks in *otu5-1* plants. (**A**,**B**) Schematic diagrams in scale showing the *MAF4* (**A**) and *MAF5* (**B**) gene structures depicting six and five amplification regions (bars), respectively, labeled 1-6 or 1-5, which were analyzed by qChIP. The gene structures of *MAF4* and *MAF5* are depicted for *FLC* in Figure 4. The corresponding primer pairs are listed in Appendix A. (**C**,**D**) The percentage qChIP amplification yields, in comparison with that amplified from input DNA, from six *MAF4* (**C**) or four to five *MAF5* (**D**) regions using various antibodies, as indicated against H3K36me3, H3K27me3, H3K9me2, and ubH2B in *otu5-1* and Col-0 seedlings. Error bars indicate standard deviations.

**Figure 6 ijms-24-06176-f006:**
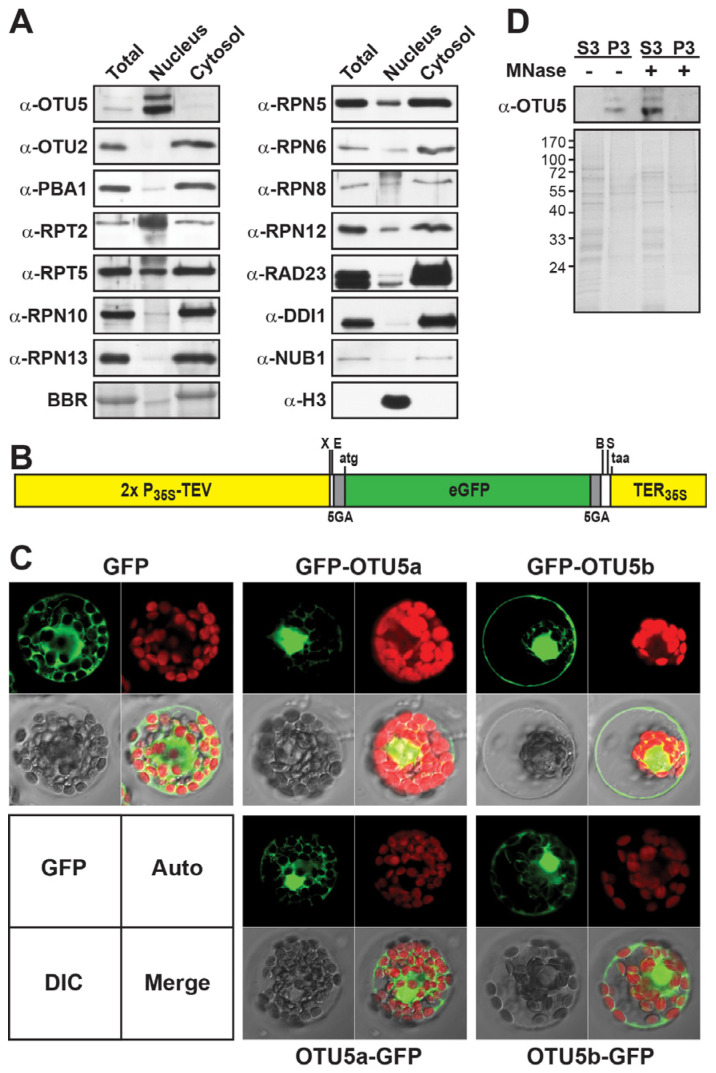
The *A. thaliana* OTU5a and OTU5b proteins are nucleus-enriched and chromatin-associated. (**A**) The relative abundance of two OTU proteins, various proteasome subunits, and ubiquitin shuttle receptors in total, nuclear, and cytosolic protein fractions prepared from 10 DAS Col-0 seedlings was examined by immunoblotting with various antisera, as indicated. Antisera against histone H3 (α-H3) were analyzed to verify the purity of the nuclear fraction. Equal amounts of proteins from each fraction were analyzed. Duplicate samples were stained with BBR to show loaded proteins. (**B**) Constructs of green fluorescence protein (GFP) fusions used for transient expression were built in p2X35S-TEV-eGFP-35ST (see Section 4), with relevant components shown schematically. The restriction sites used for N- and C-terminal fusions were *Xma*I (X)/*Eco*RI (E) and *Bam*HI (B)/*Sal*I (S), respectively. Two linker sequences encoding five GA repeats (5GA) were designed to preserve the structural integrity of the introduced fusion proteins. (**C**) Nuclear enrichment of OTU5a and OTU5b was detected by transient expression of their GFP fusions in Arabidopsis protoplasts. Representative protoplasts displayed fluorescence of N- and C-terminal GFP-fused OTU5a and OTU5b. GFP alone was analyzed as a cytosolic expression reference. The split quadruple windows (bottom left) indicate that four images shown for each panel are the same optical section examined by GFP fluorescence (GFP), autofluorescence (Auto), phase contrast (DIC), and merged images (Merge). (**D**) Nuclear OTU5 isoforms were associated with chromatin. Chromatin pellets (P3) and nucleoplasmic fractions (S3) isolated from nuclei not pretreated (MNase-) or pretreated with (MNase+) 0.2 U micrococcal nuclease were subjected to immunoblotting with α-OTU5. A duplicate sample gel was stained with BBR to show loaded proteins (bottom). The molecular weight marker sizes are indicated.

**Figure 7 ijms-24-06176-f007:**
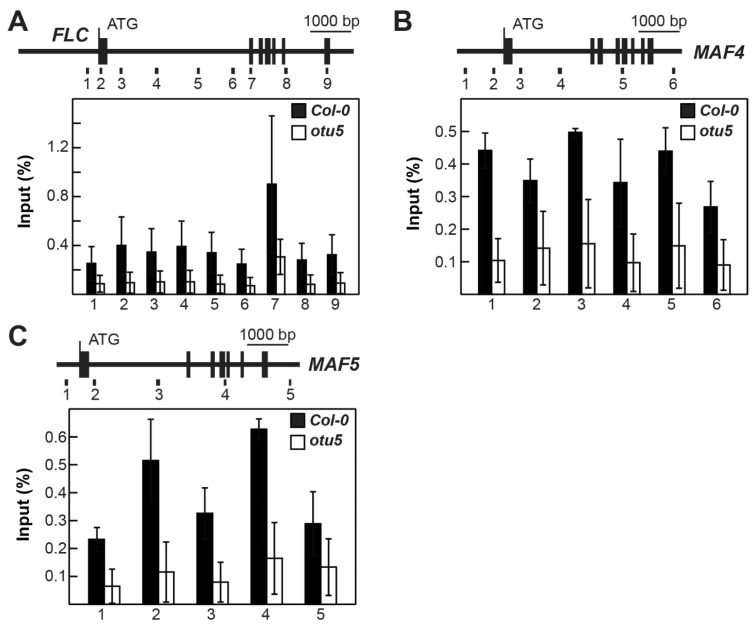
The Arabidopsis OTU5 protein associates with *FLC*, *MAF4*, and *MAF5* loci. (**A**–**C**) The association of OTU5 with various regions of *FLC* (**A**), *MAF4* (**B**), and *MAF5* (**C**) was analyzed using qCHIP. The percentage qChIP amplification yields was shown, in comparison with those amplified from input DNA, from nine, six, and five regions on *FLC*, *MAF4*, and *MAF5*, respectively. The same regions (depicted in the top panels) analyzed for various histone marks in Figure 4 and Figure 5 were analyzed using antisera against OTU5b for *otu5-1* and Col-0 seedlings. Error bars indicate standard deviations.

**Figure 8 ijms-24-06176-f008:**
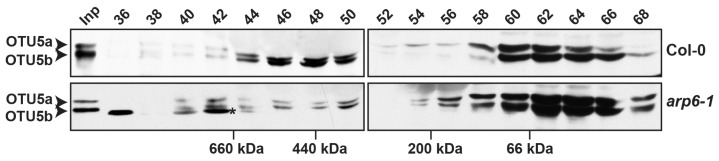
Size and abundance comparison of OTU5-associated complex(es) in wild-type and *arp6-1* seedlings. The sizes and abundance of OTU5a- and OTU5b-associated complexes were examined by gel filtration experiments using protein extracts isolated from rosette leaves of three-week-old Col-0 and *arp6-1* plants. The collected even-numbered fractions (36–68) were analyzed by SDS-PAGE followed by immunoblotting with α-OTU5 antibodies. The calibrated fraction positions of the molecular weight markers are shown at the bottom; the asterisk marks an unknown signal.

**Figure 9 ijms-24-06176-f009:**
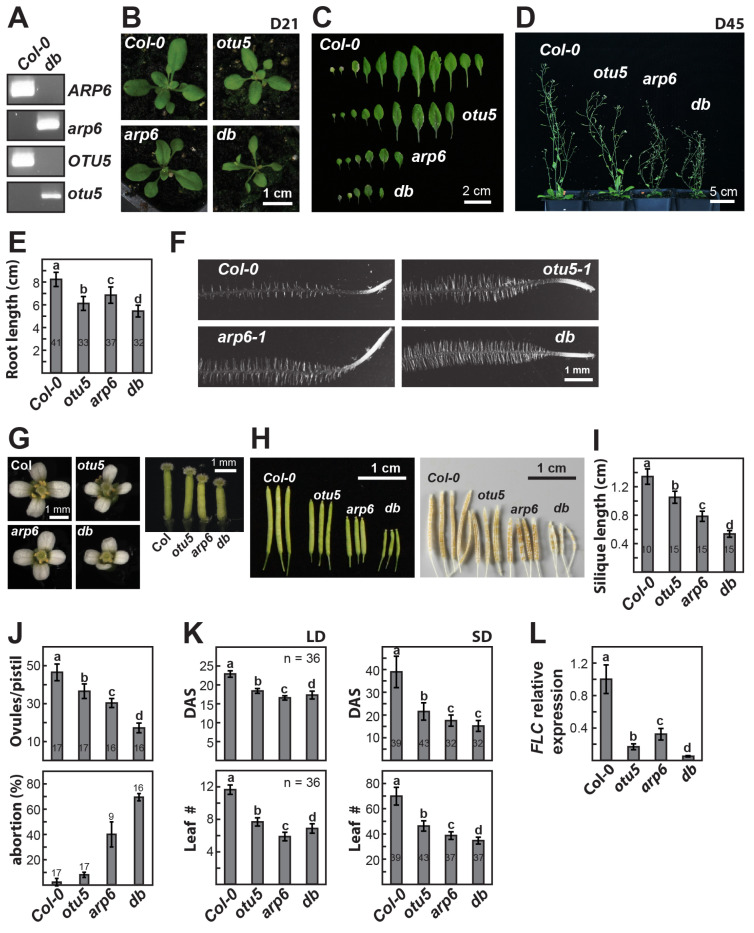
Homozygous *otu5 arp6* plants display synergistic phenotypes. (**A**) Genotyping of *arp6 otu5* plants by genomic DNA PCR. The presence of *arp6-1* (*arp6*) and *otu5-1* (*otu5*) T-DNA junctions was detected in the *otu5 arp6* double mutant (*db*), and the wild-type *ARP6* and *OTU5* fragments were only detected with Col-0. (**B**) Representative 21 DAS plants grown under long-day conditions. (**C**) Representative sets of rosette leaves at 27 DAS in order of production. (**D**) Representative 45 DAS plants grown under long-day conditions. (**E**) Average primary root lengths of seedlings at 12 DAS grown under long-day conditions (*n* = 32–41). (**F**) Representative root tips of seedlings at seven DAS grown under phosphate-replete conditions. (**G**) Representative flowers (top views, left four panels) and gynoecia/pistils (right). (**H**) Representative mature fresh-harvested (left) or alcohol-bleached siliques (right). (**I**) Averaged lengths of matured siliques. The numbers of siliques averaged are indicated. (**J**) Averaged ovule number of each silique (top) and ovule abortion rate (bottom). The numbers of siliques averaged are indicated. (**K**) Flowering times measured either by DAS (top panels) or rosette leaf number (bottom panels, Leaf #) at bolting of plants grown under long-day (LD) or short-day (SD) conditions. (**L**) Relative *FLC* expression levels examined by RT-qPCR in *otu5-1*, *arp6-1*, and *otu5 arp6* (*db*) plants compared with Col-0. The averaged relative *FLC* expression levels were determined from three biological repeats. Different letters denote significant differences by pairwise comparison using Student’s *t*-test; error bars indicate standard deviations.

**Figure 10 ijms-24-06176-f010:**
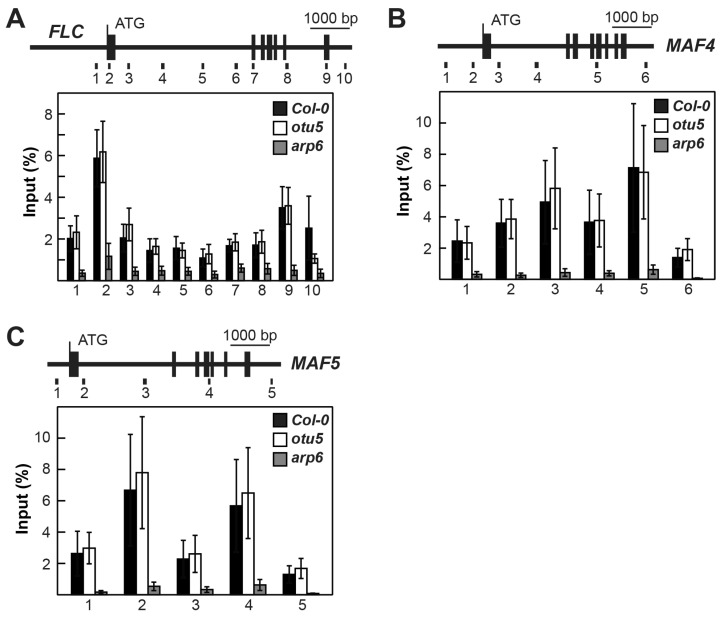
Histone H2A.Z levels on the *FLC*, *MAF4*, and *MAF5* loci. (**A**–**C**) The association of H2A.Z with *FLC* (**A**), *MAF4* (**B**), and *MAF5* (**C**) in Col-0, *arp6-1*, and *otu5-1* was analyzed using qCHIP. The percentage qChIP amplification yields from various regions depicted on the top panels (the same as those used for histone marks in Figure 4 and Figure 5) are shown relative to those amplified from the input DNA. The analysis was conducted using antisera raised against Arabidopsis H2A.Z isoforms HTA9 and HTA11. Error bars indicate standard deviations.

**Figure 11 ijms-24-06176-f011:**
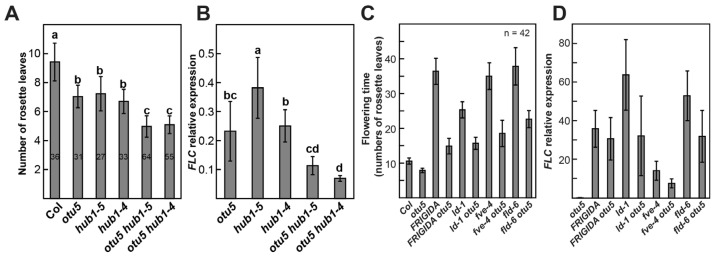
OTU5-mediated flowering suppression is HUB1-independent but partially required for autonomous pathway mutation-mediated *FLC* upregulation and late flowering. (**A**) Flowering times measured by rosette leaf number at bolting in *Col-0*, *otu5-1*, *hub1-5*, *hub1-4*, *otu5-1 hub1-5*, and *otu5-1 hub1-4* plants grown under long-day conditions; sample sizes (27–64) were indicated. (**B**) Relative *FLC* expression levels examined by RT-qPCR. (**C**) Flowering times measured by rosette leaf number at bolting under long-day conditions in *Col-0*, *otu5-1*, *Col-0*-*FRIGIDA*, various autonomous pathway mutants (*ld-1*, *fve-4*, and *fld-6*), and *otu5-1* crossed-in double mutants: *Col-0*-*FRIGIDA otu5-1*, *ld-1 otu5-1*, *fve-4 otu5-1*, and *fld-6 otu5-1*. (**D**) Relative *FLC* expression levels examined by RT-qPCR. The averaged relative *FLC* expression levels were determined from three biological repeats. Different letters denote significant differences by pairwise comparison using Student’s *t*-test. Error bars indicate standard deviations; *n* represents the sample size.

**Figure 12 ijms-24-06176-f012:**
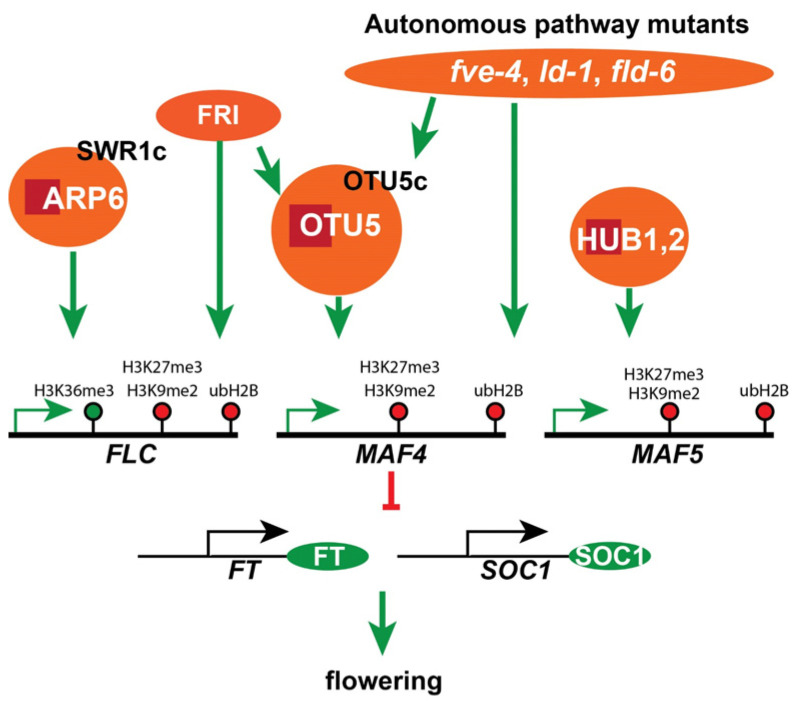
Arabidopsis OTU5 is a novel flowering suppressor. OTU5 acts, potentially in an associated protein complex (OTU5c), in parallel with SWR1c- and HUB1/2-mediated pathways to epigenetically activate *FLC* clade repressors and suppress flowering. OTU5 primarily activates *FLC* and, to a lesser extent, *MAF5*, *MAF4*, and *MAF1*, which in turn suppress major flowering initiators *FT* and *SOC1* to suppress flowering. Under unknown mechanisms, OTU5 is involved in decreasing the deposition of the activating histone mark H3K36me3 and in the upregulation of the suppressive histone marks H3K27me3 and H3K9me2 as well as ubH2B on specific loci, such as *FLC*, *MAF4*, and *MAF5*. OTU5-mediated flowering suppression is distinct from the SWR1c-mediated pathway, which activates *FLC*-related repressors through H2A.Z deposition. OTU5 also acts independently of the HUB1/2-mediated pathway, which modulates genome-wide ubH2B levels and activates a distinct set of *FLC*-related repressors through distinct activating/repressive histone marks (see Section 3 for details). OTU5 is, however, partially required for *FLC*-mediated flowering suppression in autonomous pathway mutants and *FRIGIDA*-Col. Transcription activation or flowering promotion is indicated by green arrows, and transcription suppression is denoted by a red bar.

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
