# Peer review of "The Arabidopsis Deubiquitylase OTU5 Suppresses Flowering by Histone Modification-Mediated Activation of the Major Flowering Repressors FLC, MAF4, and MAF5"

_ijms, 2023, doi:10.3390/ijms24076176_

Round 1
Reviewer 1 Report
see attachment.

Author Response
please see attached response letter to all three reviewers

Reviewer 2 Report
General comments
The manuscript entitled “The Arabidopsis Deubiquitylase OTU5 Suppresses Flowering by Histone Modification-Mediated Activation of the Major Flowering Repressors FLC, MAF4, and MAF5” is good approach encompass a better direction in flowering biology of plants for future studies.
Some shortcomings need to be addressed before this Article is recommended to be published.
The gene name should be italicized in the title and through the manuscript.
The manuscript should be checked for minor English grammar mistakes.
Abstract
Please add the objectives and methodology to this section briefly.
Introduction
This part is written but the authors and the authors tried to provide each detail in a comprehensive way. This may not be a good choice for a research article. The authors are advised to divide this part into maximum of 3 or 4 paragraphs briefly explain the significance of current work with clear objectives at the end.
Results
This section is written well but some revisions are suggested to check the results for English grammar/correction.
Discussion
This section discussed the results in relation with the previous literature, however, justification for each result should be provided.
This section should also be checked for English language and grammar correction.
Materials and Methods
The authors are advised to add the missing results and add the relevant citation.
Conclusion
The authors are advised to make an inclusive conclusions describing the main results of the current work and its significance for future works.
Author Response

(The authors gave the same response as above.)

Reviewer 3 Report
Generally, the study is fairly interesting as they demonstrate that the deubiquitylase OTU5 regulates flowering by changing the histone modifications of core MADS-box genes such as FLC, MAF4, and MAF5. They also found that OTU5 acts independently of ARP6 and HUB1 by genetic analyses.
The work consists of a large number of experiments, and the phenotypic figures are quite beautiful and well-organized. The writing is good, but further checking needs to be made to avoid any typos.
I have some questions for the authors and they may address them for improvements.
1. Figure 1A, the complementation lines seem to have a higher protein level of OTU5 than that in overexpression lines. Could the authors provide any explanation?
2. Figure 1B, why does not the complementation in CS lines work?
3. Do the authors see obvious differences between complementation and overexpression lines? In the CS lines, the plants have the normal expression of OTU5 after complementation and overexpression (Figure 1A) why do the lines show similar phenotypes with otu5?
4. Have OTU5 been tested for its enzymatic activity or is it a putative deubiquitylase and what are the targets?
5. Figures 4 and 5, the authors found that several histone methylations are also changed in mutants, as the OTU5 is an identified or a putative deubiquitylase, would the authors provide any hints for why the histone methylations are affected?
6. Figures 3 to 5 and 7, have authors done statistical analyses for the differences between WT and otu5?
7. Figure 6D, why do the S3 have a stronger signal of OTU5 after MNase treatment?
8. Figures 9E and 9F, do the pictures show the full primary roots? According to the figure, the primary root length in arp6-1 is slightly longer than that in Col-0 and all mutants have much more and longer lateral roots. Would the authors provide some explanation?
Author Response

(The authors gave the same response as above.)

Round 2
Reviewer 1 Report
Typographical and word errors are corrected. The manuscript has been improved and is suitable for publication now.
Reviewer 3 Report
I think the response and revised manuscript have already answered my concerns. I recommend accepting the nice manuscript in the current version now. Thanks very much for the authors' efforts.